



# Atmospheric circulation changes and their impact on extreme sea levels around Australia

**Frank Colberg[1], Kathleen L. McInnes[2], Julian O'Grady[2], and Ron Hoeke[2]**

[1]Australia Bureau of Meteorology, Melbourne VIC 3001, Australia
[2]Climate Science Centre, CSIRO Marine and Atmospheric Research Aspendale, Aspendale VIC 3195, Australia

**Correspondence:** Kathleen L. McInnes (kathleen.mcinnes@csiro.au)

**Abstract.** Projections of sea level rise (SLR) will lead to increasing coastal impacts during extreme sea level events globally; however, there is significant uncertainty around short-term coastal sea level variability and the attendant frequency and severity of extreme sea level events. In this study, we investigate drivers of coastal sea level variability (including extremes) around Australia by means of historical conditions as well as future changes under a high greenhouse gas emissions scenario (RCP 8.5). To do this, a multi-decade hindcast simulation is validated against tide gauge data. The role of tide–surge interaction is assessed and found to have negligible effects on storm surge characteristic heights over most of the coastline. For future projections, 20-year-long simulations are carried out over the time periods 1981–1999 and 2081–2099 using atmospheric forcing from four CMIP5 climate models. Changes in extreme sea levels are apparent, but there are large inter-model differences. On the southern mainland coast all models simulated a southward movement of the subtropical ridge which led to a small reduction in sea level extremes in the hydrodynamic simulations. Sea level changes over the Gulf of Carpentaria in the north are largest and positive during austral summer in two out of the four models. In these models, changes to the northwest monsoon appear to be the cause of the sea level response. These simulations highlight a sensitivity of this semi-enclosed gulf to changes in large-scale dynamics in this region and indicate that further assessment of the potential changes to the northwest monsoon in a larger multi-model ensemble should be investigated, together with the northwest monsoon's effect on extreme sea levels.

## 1 Introduction

Extreme sea levels (ESLs) are a significant hazard for many low-lying coastal communities (Hanson et al., 2011; Nicholls et al., 2011), and with rising global mean sea level, extreme events are expected to rise (Menéndez and Woodworth, 2010). ESLs are largely driven by storm surge superimposed on the astronomical tides (storm tides). The severity of these ESLs can be further enhanced by larger-scale atmospheric and oceanic circulation patterns that operate on seasonal to interannual timescales.

ESL hazards are typically represented as probability-based exceedance levels with associated uncertainties. These uncertainties may be significantly larger than uncertainties in projected sea level rise (SLR) itself (Wahl et al., 2017). Many studies have attempted to quantify ESL uncertainties using historical tide gauge information combined with global SLR projections (e.g. Hunter et al., 2013) or by spatially extrapolating tide gauge observations using a hydrodynamic model (e.g. Haigh et al., 2014a). In the present study, we assess the performance of a hydrodynamic model for the Australian region and examine atmospheric drivers of ESL and how they may change under future climate conditions.

A number of studies have used a similar approach, i.e. investigating ESL changes using hydrodynamic models forced by global climate models (GCMs) or regional climate models (RCMs). Lowe et al. (2009) developed projections of storm surge change for the UK using climate forcing from an 11-member perturbed physics ensemble of the Hadley Centre GCM downscaled to 25 km resolution with the RCM HadRM3 (Murphy et al., 2007) under a mid-range Special Report on Emissions Scenarios (SRES) (Nakićenović and Swart, 2000) emission scenario. Results indicated that the

changes in the 2- to 50-year storm surge height associated with projected changes in weather and storms would increase by no more than 0.09 m by 2100 anywhere around the UK coast. Sterl et al. (2009) concatenated the output from a 17-member ensemble of a mid-range SRES emissions scenario from the ECHAM5/MPI-OM climate model (Jungclaus et al., 2006) to estimate the 10 000-year return values of surge heights along the Dutch coastline. No statistically significant change in this value was projected for the 21st century because projected wind speed changes were associated with non-surge-generating southwesterlies rather than surge-conducive northerlies. Vousdoukas et al. (2016) used a hydrodynamic model to downscale storm surge changes in an eight-member ensemble of climate models under RCP 4.5 and 8.5 and found increases in storm surges over the model domain north of 50° N, whereas there was minimal to slightly negative change south of 50° N except under RCP 8.5 towards the end of the century. In southern Europe, Marcos et al. (2011) assessed changes in storm surges in the Mediterranean Sea and Atlantic Iberian coasts using climate model forcing from the ARPEGE-v3 global, spectral stretched-grid climate model under a high-, medium- and low-SRES emissions scenario (Jordà et al., 2012). Findings revealed a general decrease in both the frequency and magnitude of storm surges with up to a 0.08 m reduction in the 50-year return levels. In southern Australia, Colberg and McInnes (2012) found both positive and negative changes in 95th-percentile sea level height across the southern half of the Australian continent in surge model simulations forced by the high-SRES emission scenario of the CSIRO Mark 3.5 GCM (Gordon et al., 2010) and two simulations of the CCAM (Conformal Cubic Atmospheric Model) stretched-grid global model (McGregor and Dix, 2008). The ESL changes were small, mostly negative along the southern mainland coast but with wintertime increases over Tasmania. These resembled the changes in wind patterns to some degree, although there were large inter-model differences.

Several studies have also examined the non-linear effect of rising sea levels on tide and surge propagation. Using a global tide model, Pickering (2017) found that changes in mean high tide levels exceeded ±10 % of the SLR at approximately 10 % of coastal cities when coastlines were held fixed but a reduction in tidal range when coastlines were allowed to recede due to resulting changes in the period of oscillation. Arns et al. (2015) investigated the non-linear impact of SLR on maximum storm surge heights in the North Sea, focusing on the German Bight. They found that maximum storm surges relative to the imposed background sea levels were amplified by up to 20 % when the background mean sea levels were elevated by around 0.5 m. The positive increases in extreme water levels were caused by non-linear changes in the tidal component, which were only partially offset by a reduction in the storm surge component.

Coastal regions affected by tropical cyclones have been the focus of several recent studies. For example, Unnikrishnan et al. (2011) used RCM simulations to force a storm surge model for the Bay of Bengal and found that the combined effect of mean SLR of 4 mm yr$^{-1}$ and RCM projections for the high-emissions scenario (2071–2100) gave an increase in 1-in-100-year heights in the range of 15 %–20 % compared to the 1961–1990 baseline. For east Asia, Yasuda et al. (2014) applied a hydrodynamic model based on a 20 km resolution climate model and found that storm surge heights increased in the future for much of the coastline considered. For New York, Lin et al. (2012) investigated the change in extreme sea levels arising from hurricanes over 2081–2100 relative to 1981–2000 in four GCMs run with the SRES medium-emission scenario by generating synthetic cyclones under the background conditions provided by the GCMs. Accounting for hurricane forcing only, results differed markedly between the four climate models ranging from overall increases to decreases in storm surge level. McInnes et al. (2014, 2016a) used a synthetic cyclone technique to investigate the effect of a 10 % increase in cyclone intensity and a frequency reduction of 25 % (consistent with tropical cyclone projections for the region) on storm tides over Fiji and Samoa and found a reduction in storm tides with return periods of less than 50 years and an increase for return periods longer than 200 years.

In new studies, the contribution of waves to extreme sea levels as well as storm surge and sea level rise has also been examined. For Europe, Vousdoukas et al. (2017), using a six-member ensemble of climate models to assess changes in extreme sea levels, found that by 2100, under RCP 8.5, changes in storm surges and waves enhance the effects of SLR along the majority of northern European coasts by up to 40 %, whereas for southern Europe, decreases in storm surges and waves tend to offset the increases in extreme sea levels due to mean sea level rise. For the Mediterranean, Lionello et al. (2017) used a seven-member ensemble of regional climate model simulations under the SRES A1B scenario to examine sea level changes by 2050 and found that the positive contribution to sea level extremes of the steric (thermal expansion) component of SLR would be largely offset by the declining trend in storms and hence storm surges and waves over this time period. However, the mass addition (melting of land ice) component of SLR will likely determine an increase in water level maxima. In a global study, Vousdoukas et al. (2018) shows that under RCP 4.5 and RCP 8.5, the global average 100-year extreme sea level arising from mean sea level, tides, wind waves and storm surges is very likely to increase by 34–76 and 58–172 cm respectively between 2000 and 2100.

Numerical modelling studies of the non-linear interactions between sea level rise and cyclone-induced extreme water levels due to tides, storm surge and waves have also been undertaken. Smith et al. (2010) showed that sea level rise altered the speed of the propagation of tropical cyclone-induced storm surges on the southeastern Louisiana coast and amplified the extreme water levels under SLR although the amount of amplification varied significantly along differ-

ent parts of the coast. Hoeke et al. (2015) found that SLR reduced wave set-up and wind set-up by 10 %–20 % but increased wave energy reaching the shore by up to 200 % under cyclone conditions along the Apia, Samoa, coastline.

Australia extends from the tropics to the midlatitudes with a variety of meteorological systems responsible for extreme sea levels along its coastline (McInnes et al., 2016b). The range of weather systems, and more particularly their associated spatial scales, means that it is challenging to obtain meteorological forcing that consistently represents all weather systems responsible for sea level extremes. McInnes et al. (2009, 2012, 2013) used joint probability methods to evaluate ESLs in southeastern Australia. Haigh et al. (2014a, b) extended such modelling and analysis of ESLs to the entire Australian coast using two approaches. In Haigh et al. (2014a), the water levels arising from weather and tides were investigated over the period 1949 to 2009 using 6-hourly meteorological forcing obtained from the NCEP reanalyses, while in Haigh et al. (2014b), ESLs were simulated using a synthetic cyclone approach. As expected, extreme sea levels over the tropical northern coastlines were underestimated in the first study compared to the second one because of the low resolution of tropical cyclones in the reanalysis data set.

The present study assesses the performance of a medium-resolution barotropic hydrodynamic model for the Australian region to investigate extreme sea levels for the current climate and examines for the first time over the entire Australian coastline the potential changes in a future climate scenario in a four-member ensemble of climate model simulations. The model described by Colberg and McInnes (2012) is extended to cover the entire Australian coastline at 5 km resolution. A current climate (baseline) simulation is undertaken with tide and atmospheric forcing over the period 1981–2012 using reanalyses from the NCEP Climate Forecast System Reanalyses (CFSR) (Saha et al., 2010). The performance of the model is assessed with respect to tides, weather-driven sea levels and tide–surge interaction. Finally, changes are investigated in storm surge and seasonal sea levels around the coastline based on forcing from an ensemble of four CMIP5 models forced with the RCP 8.5 emission scenario (Taylor et al., 2012).

The paper is organised as follows. Section 2 describes the model set-up CE1, input data sets and procedure for assessing model performance. Section 3 assesses the model performance and the baseline simulations are used to investigate tide–surge interaction around the Australian coastline and the meteorological causes of ESLs. Section 4 presents the results from simulations forced by climate models, and Sect. 5 discusses the results, conclusions and further work.

## 2 Model description and method

### 2.1 Model configuration

As with Colberg and McInnes (2012), the model used in this study is the Rutgers version of the Regional Ocean Modeling System (ROMS) (Shchepetkin and McWilliams, 2005) configured to run in barotropic or "depth-averaged" mode. The model grid spans the region shown in Fig. 1 at 5 km resolution. Bathymetry for the model is obtained from the $1' \times 1'$ resolution General Bathymetric Chart of the Oceans data set (GEBCO; Jakobsson et al., 2008).

For simulations including tides, the tidal currents and heights were derived from the TPXO7.2 global model (Egbert et al., 1994; Egbert and Erofeeva, 2002) and applied to the open model boundaries. TPXO7.2 best fits (in a least-squares sense) the Laplace tidal equations and along track-averaged data from the TOPEX/Poseidon and Jason altimetry missions, obtained with OTIS (Oregon State University Tidal Inversion Software). Eight primary tidal constituents (M2, S2, N2, K2, K1, O1, P1, Q1) are provided on a 0.25° of a degree resolution full global grid. A combination of Flather–Chapman boundary conditions was used in applying the tidal forcing (Flather, 1976; Chapman, 1985). The Flather condition was applied to the normal component of the barotropic velocity and radiates deviations from the values at exterior grid points out of the model domain at the speed of the external gravity waves. The corresponding Chapman condition for surface elevation assumes all outgoing signals leave at the shallow-water wave speed. Meteorological forcing is discussed in the next section.

### 2.2 Baseline experiment

In the first part of the study, we assess the ability of the Australia-wide ROMS model to simulate historical tides and meteorologically driven water levels. The model experiments performed are also used to investigate non-linear tide–surge interactions as well as the meteorological causes of extreme sea levels around the Australian coastline. Three baseline experiments are run over the period 1981–2012. The first experiment (B-TM) includes tidal and meteorological forcing, the second (B-T) tide forcing only and the third (B-M) meteorological forcing only. Meteorological forcing for these experiments is obtained from the CFSR data set (Saha et al., 2010, 2014), which provides meteorological variables across the globe at hourly temporal resolution and approximately 38 km spatial resolution from 1979 to 2012.

### 2.3 Climate change experiments

Finally, a set of simulations with meteorological forcing from four GCMs from the fifth phase of the Coupled Model Intercomparison Project (CMIP5; Taylor et al., 2012) is undertaken to assess how climate change will affect sea levels around the Australian coast. Sub-daily atmospheric forcing

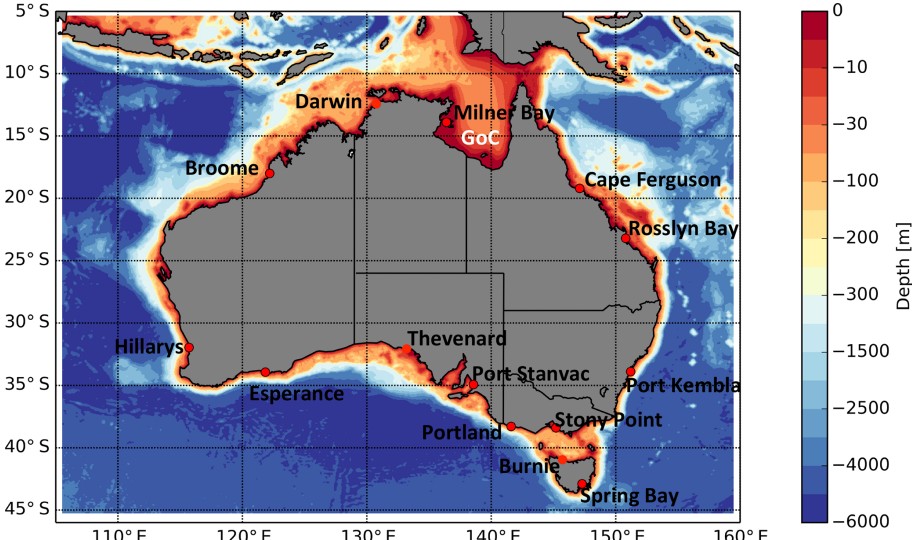

**Figure 1.** The region covered by the hydrodynamic model and bathymetric depth is shown. Red dots mark the locations of the tide gauges used for validation of baseline simulations listed in Table 2. The Gulf of Carpentaria is indicated by GoC.

data are a necessary requirement for the climate simulations undertaken in this study. CMIP5 models that are providing the output frequency as needed only do so on selected 20-year time slices from the late 20th and 21st century, thereby limiting the available data to force the hydrodynamic model. The 20-year time slices are deemed adequate for assessing how large-scale circulation changes will affect the drivers of ESLs around much of the Australian coast where seasonally varying weather systems are a major cause of extreme sea levels. This may not be the case in areas where El Niño–Southern Oscillation (ENSO) is associated with significant interannual sea level variability. ENSO-induced sea level variability is confined largely to Australia's northern coastline westward from the Gulf of Carpentaria and is minimal on the east and southern coasts (e.g. see Fig. 2a in McInnes et al., 2016b). Four models were chosen by subjectively evaluating performance metrics reported by Hemer and Trenham (2016) as well as computational considerations and data availability (see Table 1 for details).

Tides were not included in the simulations forced by climate models. This was primarily because of the large computational overhead required to undertake two simulations for each time slice (current and future) consisting of one simulation with tides only and one with tides and atmospheric forcing in order to calculate non-tidal residuals. As will be discussed in Sect. 3.3, the decision to omit tidal forcing from the climate runs is somewhat justified because although non-linear tide–surge interaction is evident around parts of Australia and may impact substantially on an individual surge event, it does not change the surge statistics over a period of years to decades significantly for most locations around Australia (Williams et al., 2016), which is the main focus of the

experiments. In the following we refer to the climate change simulations as CC (see also Table 1).

## 3 Baseline results and model performance

Here we assess the baseline experiments (forced by CFSR and/or tides) in terms of how well the model-generated sea levels compared with observations. In the first subsection, we address the contribution of seasonal and interannual variability in sea level in the modelled and observed data. The following subsections examine the performance of the model in representing astronomical tides, the high-frequency variability in sea levels including extremes, and the meteorological drivers of ESLs around the coast. Finally, we examine tide–surge interaction.

The model is assessed against hourly tide measurements from 14 high-quality tide gauges from the Australian Base Line Sea Level Monitoring Network with data available from 1993 to 2012 (Fig. 1). We decompose both the tide gauge measurements and the simulated sea levels at corresponding model grid points in the B-TM simulation into components consisting of the (a) seasonal and interannual variability, (b) the tidal signal and (c) the residual signal (the remaining signal after the removal of the seasonal and tidal components from the total sea level) using the approach of Haigh et al. (2014a). In order to facilitate a fair comparison between modelled and observed time series, we apply the same methodology to both. Firstly, sea levels are linearly detrended at each station. The seasonal and interannual component is then derived by applying a 30-day running mean to the detrended time series. The running mean is removed in the next step and a harmonic tidal analysis is carried out

**Table 1.** Summary of model experiments carried out. The spatial and temporal resolutions refer to the source of the atmospheric forcing applied to the ROMS model.

| | ROMS model experiments | Time period | | Atmospheric forcing | Emission scenario | Spatial resolution (° lat × ° long) | Temporal resolution (h) | Tide |
|---|---|---|---|---|---|---|---|---|
| | | Historical | Future | | | | | |
| Baseline | | | | | | | | |
| B-TM | Tide + meteorology | 1981–2012 | | CFSR | | 0.3° × 0.3° | Hourly | Yes |
| B-T | Tide only | 1981–2012 | | None | | | | Yes |
| B-M | Meteorology only | 1981–2012 | | CFSR | | 0.3° × 0.3° | Hourly | No |
| Climate change | | | | | | | | |
| CC-A | ACCESS1.0 | 1980–1999 | 2080–2099 | ACCESS1.0 | RCP 8.5 | 1.9° × 1.2° | 3-hourly | No |
| CC-H | HadGEM-ES | 1980–1999 | 2080–2099 | HadGEM-ES | RCP 8.5 | 1.9° × 1.2° | 3-hourly | No |
| CC-I | INMCM4 | 1980–1999 | 2080–2099 | INMCM4 | RCP 8.5 | 2.0° × 1.5° | 3-hourly | No |
| CC-C | CNRM-CM5 | 1980–1999 | 2080–2099 | CNRM-CM5 | RCP 8.5 | 1.4° × 1.4° | 3-hourly | No |

using T TIDE (Pawlowicz et al., 2002). This yields the tidal signal. Removing the tidal signal from the time series gives the residuals, which include the storm surge signal.

These component time series, as well as the total sea level, are compared by means of root mean square errors (RMSEs), the mean difference in standard deviation between observations and simulation (STDEs) and linear correlations between the modelled and observed time series over the period from 1993 to 2012 (the shorter assessment period is determined by the availability of tide gauge data at the selected sites). In addition, a 1-day running mean filter was applied to the de-tided modelled and observed sea levels for the locations of Darwin and Broome because these locations display high-frequency intra-daily to daily variability in sea surface height after applying the filtering techniques described above. This variability may be a consequence of the large tidal signal in the area propagating over a fairly shallow and wide shelf. The nature of the high-frequency variability is such that at times it would mask surge events related to atmospheric weather patterns.

### 3.1 Seasonal and interannual variations in sea level

Table 2 compares the differences between the seasonal signal in the observations and the model via RMSE, STDE and correlation coefficients. For most of the coastline, the RMSE values are 0.07 m or less with lowest values along the southeast coast. Higher values of RMSE occur on the northern and western coastline from Milner Bay (0.15 m) to Hillarys (0.10 m). Similarly, STDE indicate that the model underestimates the seasonal component by a larger amount in these locations. The reason for the poorer model performance in these locations may be attributed to seasonal and interannual variations since these regions feature a relatively large steric component, which is not simulated by barotropic models (Haigh et al., 2014a).

At Milner Bay, a large seasonal cycle in sea level occurs in part due to the transition from the prevailing northwesterly winds during the December to April monsoon to the dry season southeasterly trade winds from May to November (Oliver and Thompson, 2011; Green et al., 2010) and steric effects from seasonal variations in ocean temperature and salinity. Variations in barotropic and steric sea level components are approximately in phase, are at a maximum in January and are highest in the southeast of the Gulf of Carpentaria (Forbes and Church, 1983).

The range of the seasonal signal from tide gauge measurements for Milner Bay is here estimated to be 0.67 m. This is lower than the range of approximately 0.8 m reported in Tregoning et al. (2008) based on 5 years of data, and the difference may be a result of interannual variations in the seasonal cycle in the longer record that is analysed here. The range of the seasonal signal in the barotropic model is 0.27 m, also smaller than the barotropic range of 0.4 m estimated by Tregoning et al. (2008). Nevertheless, the results highlight that the steric component contributes to about half of the seasonal variation in sea levels in the Gulf of Carpentaria.

A relatively large steric component is also present in the seasonal signal from Darwin to Hillarys, and this is related to seasonal variations in the strength of the southward flowing Leeuwin Current, which is weakest in October to March as it flows against maximum southerly winds and is strongest between April and August when southerly winds are weaker (Godfrey and Ridgway, 1985). This produces an annual cycle in sea levels at Hillarys of about 0.22 m with maximum levels occurring in May–June and minimum levels in October–November (Pattiaratchi and Eliot, 2008). The range of the seasonal signal from the Hillarys tide gauge is estimated here to be 0.34 m, whereas in the model it is 0.09 m, the difference being of a similar order to the steric effect, which is not captured by the model.

**Table 2.** Root mean square errors (RMSEs), mean standard deviation errors (STDEs) and correlation coefficients for the astronomical tide, residual, seasonal signal and total water levels for the period 1993 to 2012 (except for Port Stanvac, which is over the period 1993–2009). For sites marked with (*) a 24 h running mean was applied to both the de-tided observations and model simulations to remove noise arising from the de-tiding process that was most pronounced at these locations.

| Site name | RMSE | | | | STDE | | | | Correlation | | | |
|---|---|---|---|---|---|---|---|---|---|---|---|---|
| | Season. | Tide | Resid. | Total | Season. | Tide | Resid. | Total | Season. | Tide | Resid. | Total |
| Spring Bay | 0.04 | 0.07 | 0.06 | 0.11 | −0.01 | 0.06 | 0.00 | 0.05 | 0.62 | 0.99 | 0.85 | 0.95 |
| Port Kembla | 0.05 | 0.05 | 0.06 | 0.13 | −0.03 | 0.03 | −0.02 | 0.03 | 0.39 | 0.99 | 0.56 | 0.95 |
| Rosslyn Bay* | 0.06 | 0.26 | 0.05 | 0.34 | −0.05 | −0.20 | −0.03 | −0.20 | 0.75 | 0.99 | 0.70 | 0.96 |
| Cape Ferguson* | 0.07 | 0.17 | 0.06 | 0.25 | −0.06 | 0.10 | −0.05 | 0.09 | 0.77 | 0.98 | 0.73 | 0.95 |
| Milner Bay | 0.15 | 0.15 | 0.11 | 0.23 | −0.14 | −0.09 | −0.08 | −0.15 | 0.91 | 0.90 | 0.78 | 0.85 |
| Darwin* | 0.10 | 0.30 | 0.06 | 0.42 | −0.09 | −0.02 | −0.04 | −0.03 | 0.73 | 0.98 | 0.55 | 0.97 |
| Broome* | 0.08 | 0.40 | 0.07 | 0.63 | −0.07 | −0.08 | −0.03 | −0.07 | 0.74 | 0.98 | 0.39 | 0.95 |
| Hillarys | 0.10 | 0.04 | 0.08 | 0.13 | −0.08 | −0.01 | −0.06 | −0.06 | 0.53 | 0.97 | 0.80 | 0.84 |
| Esperance | 0.07 | 0.08 | 0.08 | 0.13 | −0.05 | −0.03 | −0.04 | −0.06 | 0.57 | 0.93 | 0.81 | 0.87 |
| Thevenard | 0.07 | 0.21 | 0.11 | 0.25 | −0.04 | −0.12 | −0.06 | −0.14 | 0.62 | 0.87 | 0.84 | 0.85 |
| Port Stanvac | 0.07 | 0.37 | 0.10 | 0.39 | −0.04 | −0.19 | −0.06 | −0.20 | 0.75 | 0.67 | 0.88 | 0.70 |
| Portland | 0.06 | 0.06 | 0.06 | 0.10 | −0.03 | −0.01 | −0.02 | −0.02 | 0.66 | 0.96 | 0.86 | 0.92 |
| Stony Point | 0.05 | 0.15 | 0.07 | 0.20 | −0.02 | −0.07 | −0.03 | −0.08 | 0.66 | 0.98 | 0.84 | 0.96 |
| Burnie | 0.05 | 0.13 | 0.06 | 0.25 | −0.01 | −0.02 | −0.02 | −0.02 | 0.43 | 0.99 | 0.77 | 0.96 |
| Average | 0.07 | 0.17 | 0.07 | 0.25 | −0.05 | −0.05 | −0.04 | −0.06 | 0.65 | 0.94 | 0.74 | 0.91 |

## 3.2 Tides

A comparison of the amplitudes of the eight major tidal constituents derived from the measured and modelled sea levels over 1993–2012 is presented in Fig. 2 for each of the tide gauge locations. For most locations there is reasonably good agreement between constituents estimates from model and observations. The largest differences in the M2 and S2 constituents occur along the south coast at Thevenard and Port Stanvac. At Port Stanvac in particular, this may be related to poor resolution of tidal waves propagating into the Gulf of St. Vincent. Milner Bay in the Gulf of Carpentaria is also showing poor agreement, with the leading O1 and K1 constituents largely underestimated by the model. The RMSE values in Table 2 also reflect larger differences and lowest correlations at Port Stanvac and Thevenard. Locations with large tidal amplitudes such as Broome and Darwin display the largest RMSE errors (30 and 40 cm respectively). On average RMSE, STDE and correlation across all locations is 0.17 m, −0.05 m and 0.94 respectively, indicating generally good model skill overall.

## 3.3 Sea level residuals

The sea level residuals, obtained after removal of the tides and seasonal signal, are indicative of short-term fluctuations such as storm surge. Table 2 shows error statistics for the sea level residuals over the period 1993 to 2009, and in Fig. 3 data are plotted for selected sites for the year 1997. This particular year is selected because it contained examples of storm surges at each of the tide gauge locations across the

Australian region. The lowest RMSE errors of around 0.06 m are generally located along the east coast and within Bass Strait. The largest RMSE errors of 0.11 m are found at Milner Bay in the Gulf of Carpentaria and at Thevenard and Port Stanvac along the south coast. Correlations are highest at gauges across the south coast stretching from Hillarys to Spring Bay, with values exceeding 0.8 at all locations except Burnie, where a slightly lower correlation of 0.77 was found. Correlations are lowest in macro-tidal areas with large shelves and/or complex bathymetry, with the lowest values of 0.55 and 0.39 at Darwin and Broome respectively. The poorer performance in these areas is further demonstrated using quantile–quantile plots shown in Fig. 4. It can be seen that the ESLs tend to be more systematically underestimated along this coastline than in the southern midlatitudes. For example, at Milner Bay, the 99.9th percentile values are underestimated by approximately 0.5 m. At Port Stanvac, the underestimation of the high percentiles is likely a result of the 5 km grid spacing of the model inadequately resolving the Gulf of St. Vincent in which Port Stanvac is located.

To provide further insights into the type and scale of the synoptic weather systems responsible for the storm surge events identified by arrows in Fig. 3 (note that for Burnie, the synoptic map for Portland applies), Fig. 5 presents the mean sea level pressure (MSLP) and 10 m wind vectors at the time of the peak sea levels. At Spring Bay, the peak residual of 0.4 m on 8 July 1997 is associated with the passage of a frontal trough that brings low pressure and southwesterly winds along the eastern Tasmanian coast (Fig. 5a). McInnes et al. (2012) found that daily maximum sea levels at Spring Bay were highly correlated with those in Hobart ($r = 0.98$)

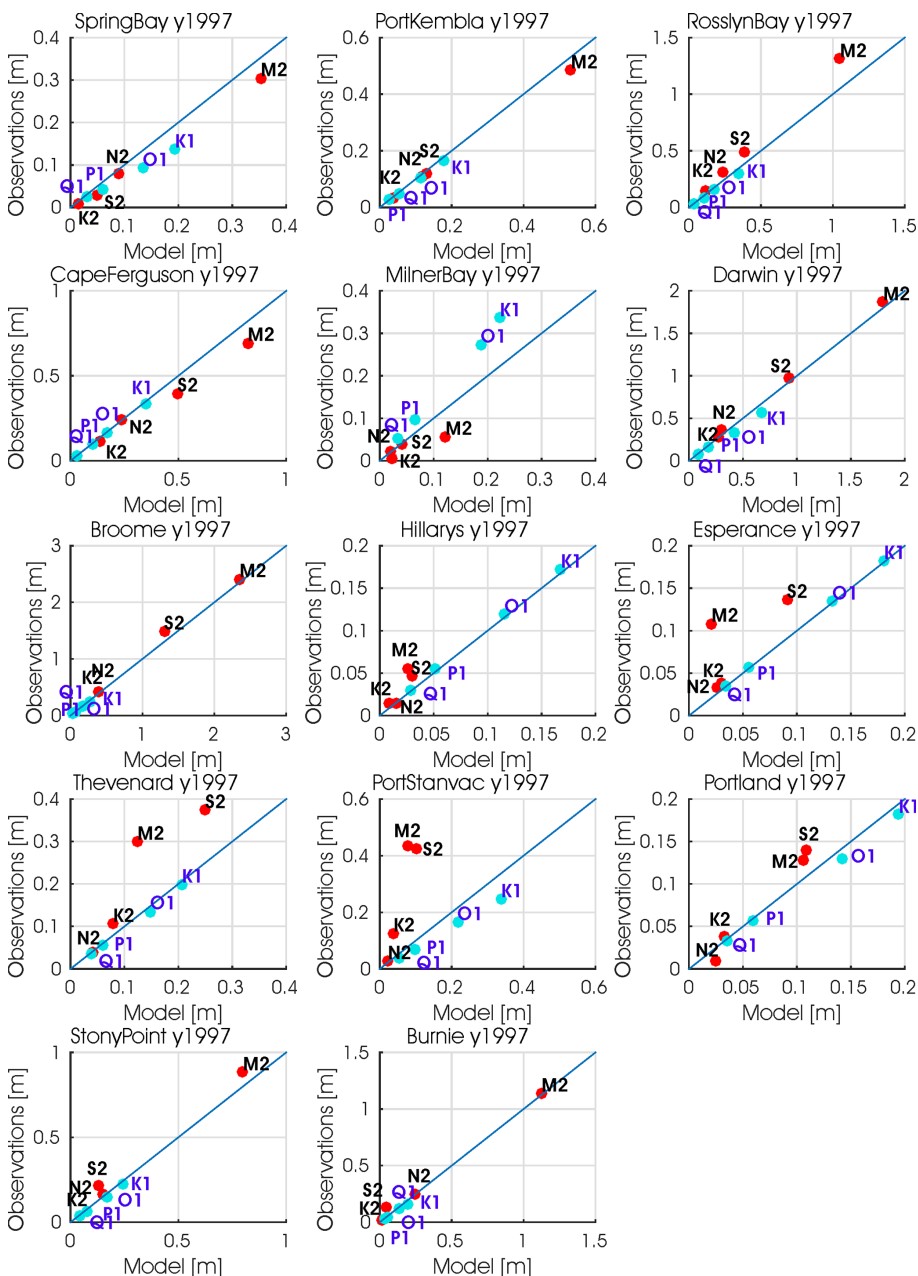

**Figure 2.** Comparison of the eight major tidal constituents estimated from observations and modelled for the tide gauge locations shown in Fig. 1. Red (blue) dots denote the semi-diurnal (diurnal) tidal constituents respectively.

and Portland ($r = 0.80$), indicating the strong influence of frontal systems on sea level extremes in this part of the country. Indeed relative peaks in residuals are evident at other south mainland coast stations for this event (Fig. 5g–j).

At Port Kembla a relative peak in residual sea level of 0.3 m at around 10 May 1997 is the result of an east coast low that brings southeasterly winds to the coast. These systems are the cause of the majority of elevated sea level events along this coastline (McInnes and Hubbert, 2001). A tropical cyclone off the northeast coast around 9 March (Figs. 3c and 5c) and in the Gulf of Carpentaria on 28 December are responsible for sea level residuals of up to 0.4 at Rosslyn Bay and 1.0 m at Milner Bay respectively (Figs. 3d and 5d). A second residual peak at Rosslyn Bay of up to 0.4 m around 13 May was not captured by the model.

The cause of this peak in the observations is not easily explained by the synoptic winds and SLP (sea level pressure) fields. However, some evidence, as described next, points towards this peak being generated by a coastally trapped wave (CTW). Coastally trapped waves travel anticlockwise around

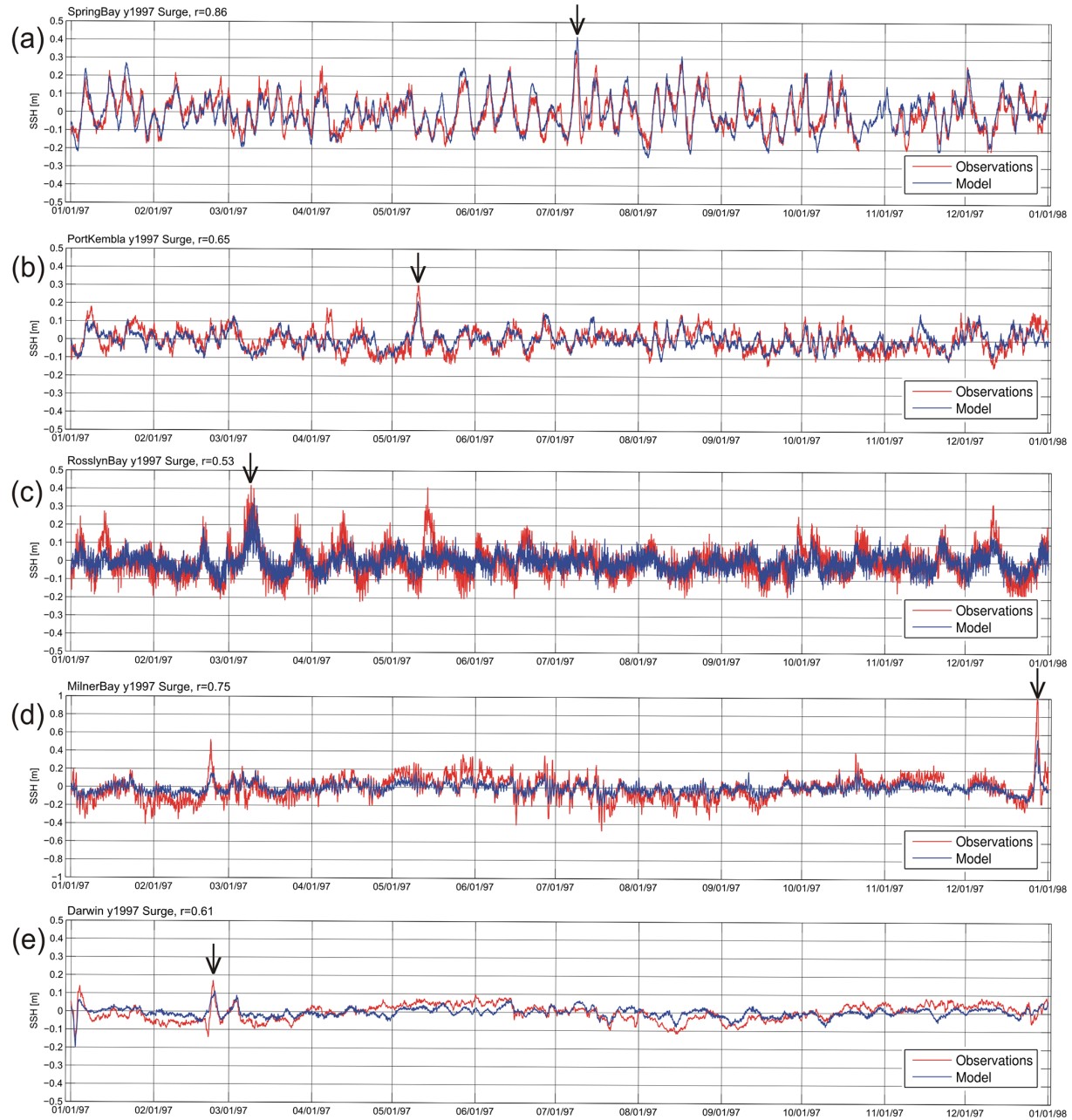

**Figure 3.**

Australia with speeds between 2 and $4\,\mathrm{m\,s^{-1}}$ and amplitudes of the order of 0.25 m (Woodham et al., 2013). On 10 May a coastal low produced a surge in Port Kembla that may have excited such a CTW. The timing and measured elevation height for the peak at Rosslyn Bay matches well with theoretical values of a passing CTW. The barotropic hydrodynamic model used in this study does not allow higher-order (baroclinic) modes of CTW to exist, and this may contribute to the failure of the model to capture this extreme sea level. Also unresolved bathymetric features over the Great Barrier

Reef may alter the modelled sea surface height signal at this location.

At Darwin, a small relative peak of about 0.2 m around 22 February is associated with a burst of northwest monsoon winds (Figs. 3e and 5e). At this time sea levels are also elevated to 0.5 m at Milner Bay (Fig. 3d) by the northwesterly winds that are also directed into the Gulf of Carpentaria. At Hillarys, a sea level peak around 18 May is associated with a low-pressure system off the southwest of the continent directing northwesterly flow onto the southwest coast. The final sequence of panels (Fig. 5g–i) shows the passage

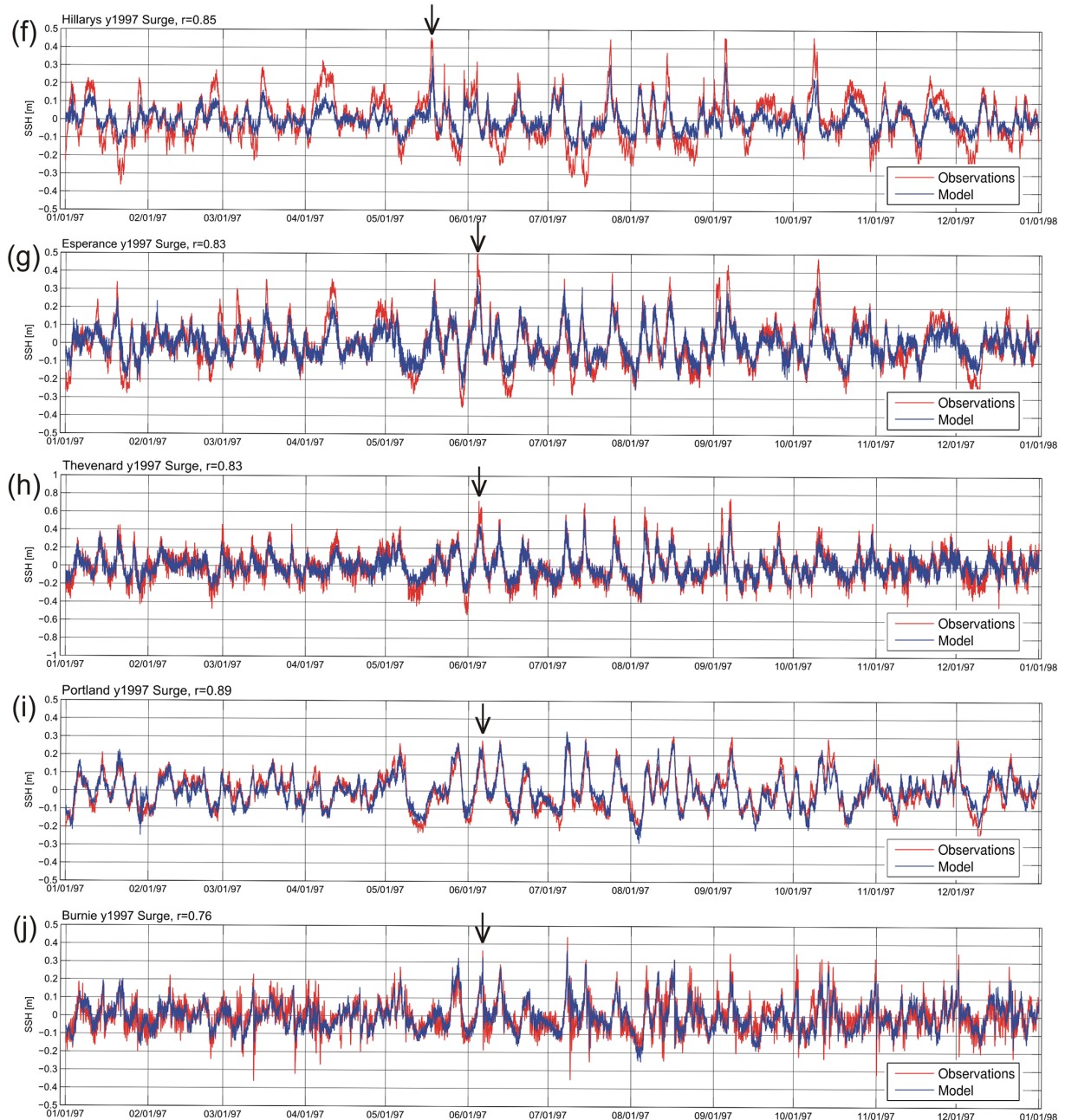

**Figure 3.** Comparison of sea level residuals from tide gauge observations (red) and baseline model experiment (B-TM) for 1997. Black arrows indicate storm surge events discussed in the text. SSH: sea surface height.

of a cold front that travels from west to east bringing south-westerly winds to the south coast of Australia and producing elevated sea levels in Esperance on 4 June (Figs. 3g and 5g), Thevenard on 5 June (Figs. 3h and 5h), and Portland 5 and Burnie on 6 June (Figs. 3i–j, 5i). Events of this type have been discussed in previous studies such as McInnes and Hubbert (2003) and McInnes et al. (2009).

### 3.4 Tide–surge interaction

Understanding tide–surge interaction is important since it can alter the timing, severity and intensity of storm surges 10 (Olbert et al., 2013; Haigh et al., 2014b; Antony and Unnikrishnan, 2013). In the context of the present study, a better understanding of the potential non-linear interaction between tides and surges contributes to an understanding of the uncertainty associated with the CMIP5-forced ocean model 15 simulations.

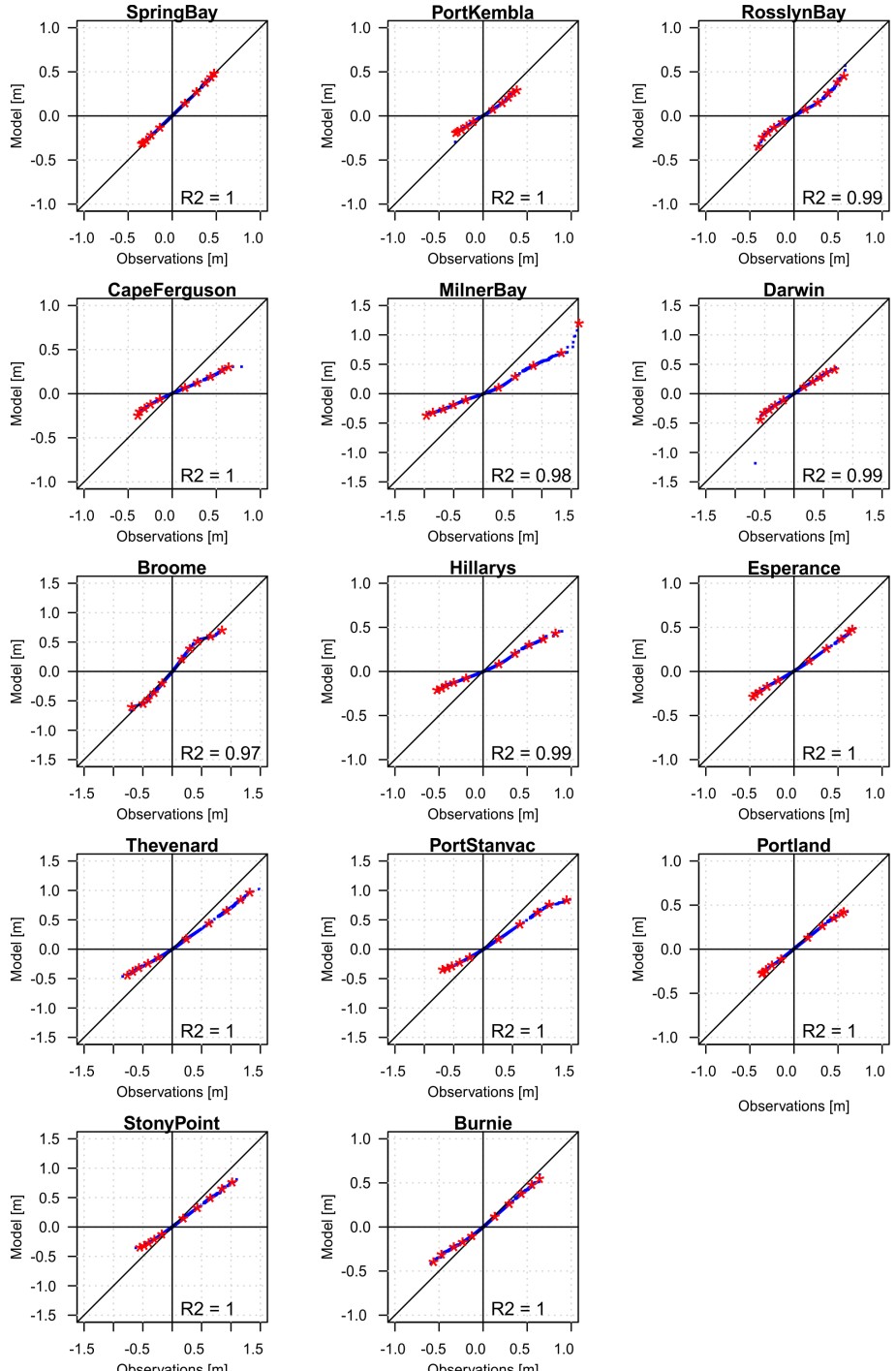

**Figure 4.** Quantile–quantile plots showing model-derived residuals vs. residuals from observations. The 0.01, 0.1, 1.0, 10, 90, 99, 99.9 and 99.99th percentiles are highlighted in red.

Tide–surge interaction has been studied previously for parts of the Australian coast. In Bass Strait, the occurrence of strong westerly winds leads to a phase shift in the timing of the surge (McInnes and Hubbert, 2003; Wijeratne et al., 2012). On the northern shelf, the combination of strong trop-

ical cyclone winds together with tides alters the amplitude of the water column (Haigh et al., 2014b). Both of these observed effects are in line with the notion of Rossiter (1961) that the interaction of tides and surges is one of mutual alteration. Simply put, depending on the size of the tide and the

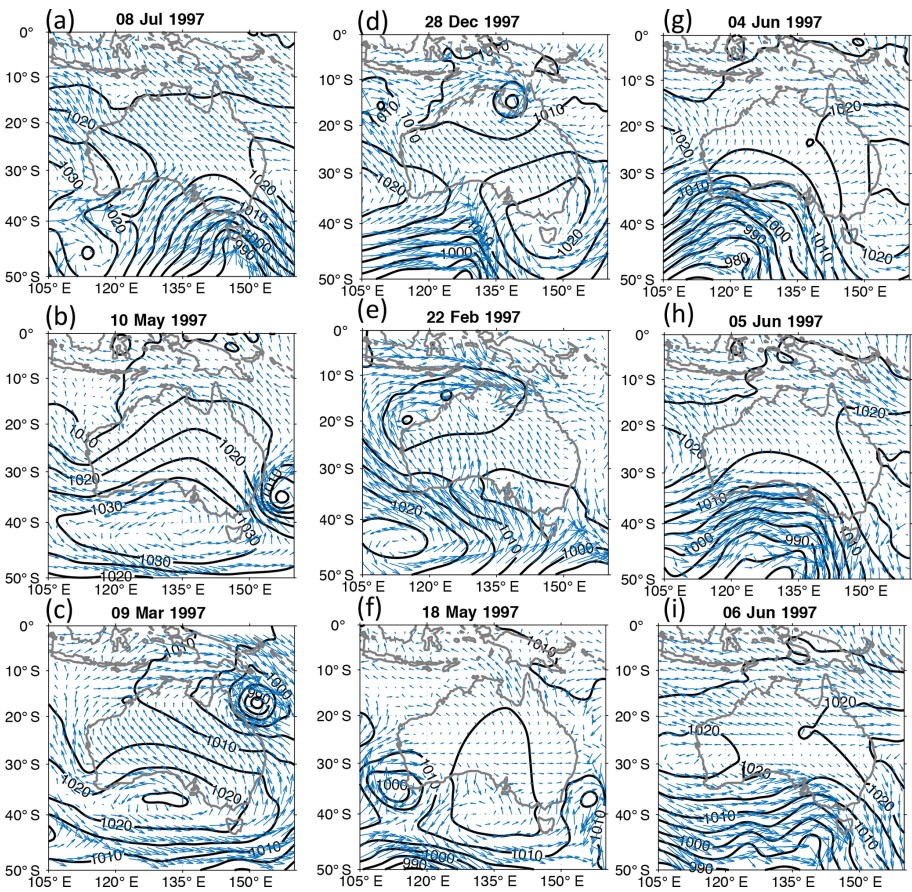

**Figure 5.** Mean sea level pressure and surface winds from CFSR reanalyses associated with the storm surge events indicated with black arrows in Fig. 3. Note, Fig. 3i and j both relate to the same synoptic pattern as Fig. 5i.

water depth the presence of tides alters the generation of the surge signal because the wind is more effective at creating a surge over lower sea levels. They conclude, therefore, that surges produced during low tide are generally larger (Horsburgh and Wilson, 2007) than those produced during high tides. Furthermore, since the tide and surge signals propagate as shallow water waves the presence of a surge increases the speed of the tidal wave so that the high tide arrives sooner than predicted. Therefore, when predicted tides are removed from tide gauge observations, the residuals can contain variations that are not driven by meteorological effects (e.g. McInnes and Hubbert, 2003).

To examine tide–surge interaction, sea level components ($\zeta$) from the three baseline simulations are analysed (see Table 1). The first is forced by meteorology (B-M, i.e. atmospheric winds and pressure only) yielding surge only, $\zeta_M$; the second (B-T) is forced by tides only, $\zeta_T$; and the third (B-TM) combines tide and meteorological forcing, $\zeta_{TM}$. Subtracting the $\zeta_T$ from the $\zeta_{TM}$ yields a time series of residuals $\zeta_R$. By definition, differences between the time series of residuals and surges (i.e. $\zeta_R$ and $\zeta_M$) are a result of tide–surge interaction.

The potential amplitude changes arising from tide–surge interactions around Australia are first examined by selecting the four largest surges and the four largest residuals (separated by a 3-day window) per year from the 20-year $\zeta_M$ and $\zeta_R$ time series respectively and ranking the values (Fig. 6). Although ranking of events removes the one-to-one relationship between the events in the surge and residual time series, it clarifies the relationship between the two. Figure 6 suggests the relationship between the surges and residuals (red points and axes on top and right) are close to 1, indicating that across the population of extremes, the height of the surge is not systematically affected by the presence of tides in B-TM. Exceptions are Broome, where the largest residuals (those greater than 0.6 m) are higher than the equivalent surges, and Darwin and Burnie, where residuals tend to be consistently higher than the surges by about 1–2 cm.

To examine the effect of non-linear interaction on the timing of the surge maximum, we also examine the total water level at the time of the four largest annual maxima from the $\zeta_R$ and $\zeta_M$. In order to do so we add the predicted tide height to the surge and residuals at the times that the respective peaks occurred and again ranked the two groups and plot

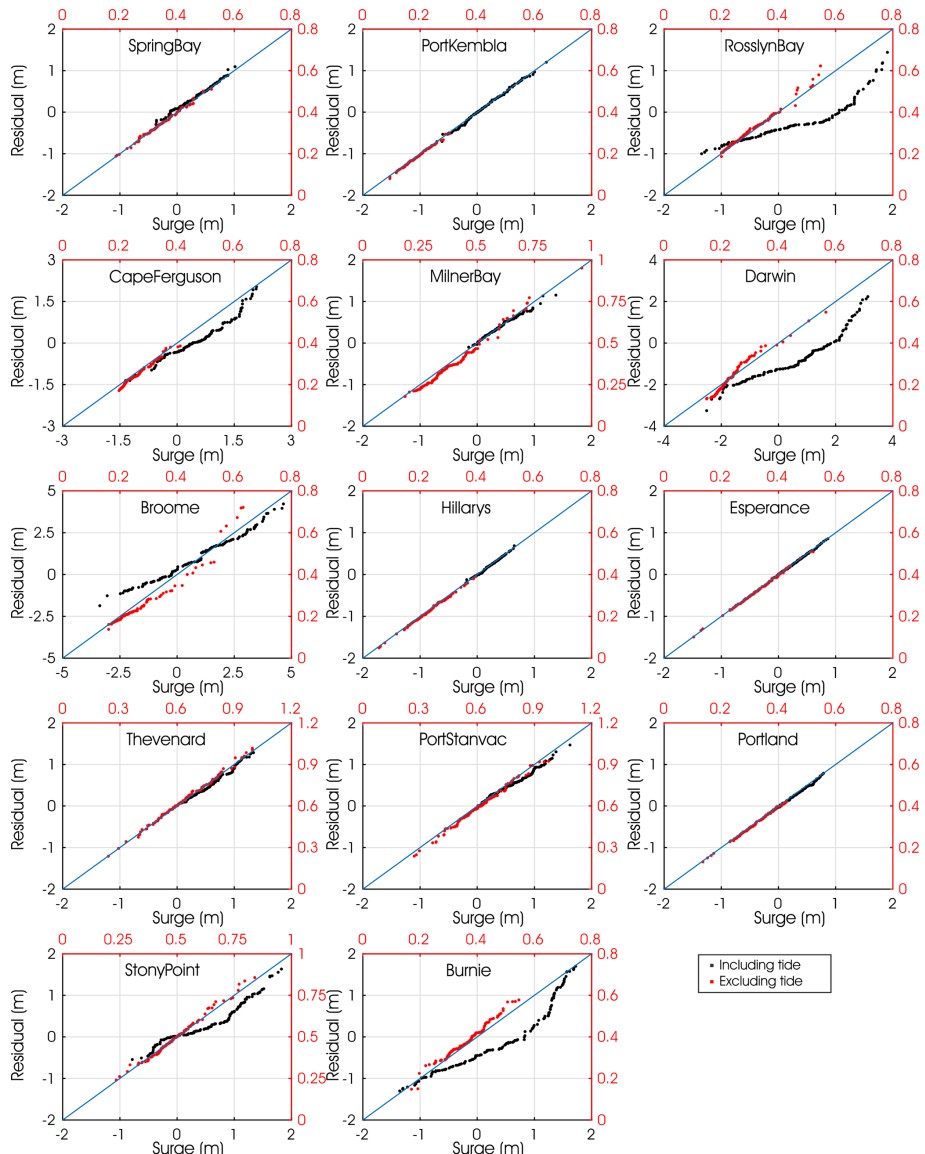

**Figure 6.** Regression plots between the four largest modelled surges and residuals per year. Units are in metres. Surges are obtained by running the ocean model with atmospheric forcing only (B-M). Residuals are obtained by subtracting tidal forced simulation (B-T) from atmospheric and tidal forced simulation (B-TM). Black: tidal elevation added at times of maximum surge/residual. Red: tidal elevation omitted at times of maximum surge/residual. Figure suggests (1) that tides can affect total sea level for some stations as maximum residuals tend to occur during low tide (black scale) and (2) that the surges and residuals are of similar order of magnitude (regression is close to 1) and are hence only affected marginally by the presence of tidal forcing (red scale).

their relationship (black points and bottom and left axes in Fig. 6). In this case near one-to-one relationships are now only seen for 8 of the 14 stations. Tide–surge interaction is evident for Cape Ferguson, Rosslyn Bay, Broome, Darwin, Burnie and Stony Point. With the exception of Broome, the interaction is such that the total sea level at the times of the maximum $\zeta_R$ is smaller than the total sea level at times of maximum $\zeta_M$. In other words when tides are included in the model simulations, the interaction between tides and surges causes the maximum sea levels to occur during lower tides.

The density distribution of the tides at the time of the four largest surges and residuals (not shown) indicates that the reason for the difference is that maximum residuals tend to occur on low waters for these locations. This "phase locking" phenomenon may occur because the presence of a surge increases the water depth and this changes the speed of the tidal wave due to the reduced bottom friction (e.g. Arns et al., 2015). As shown by Horsburgh and Wilson (2007) in observations, a first-order effect of this is that the peak surge occurs before the maximum water level due to tides only.

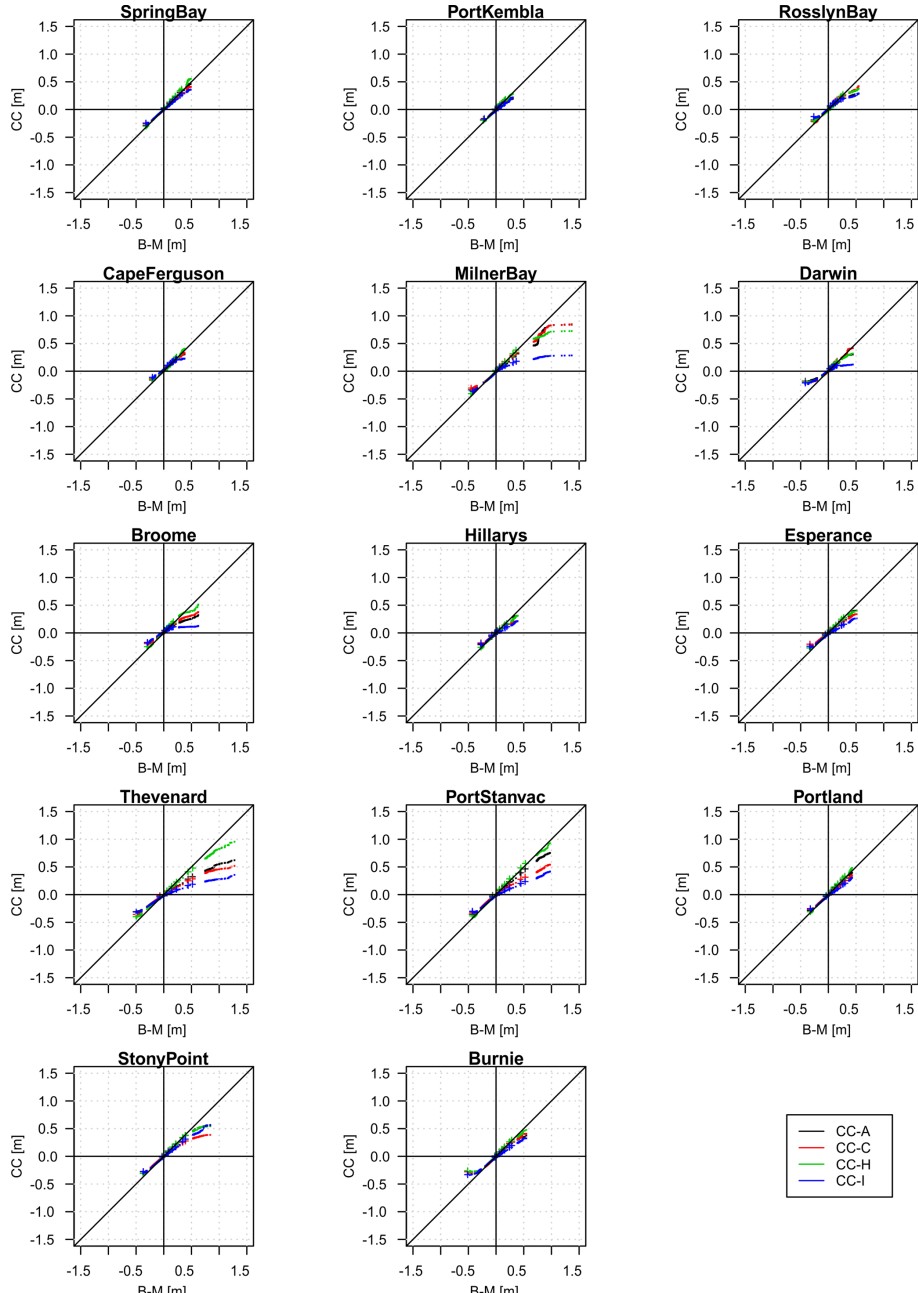

**Figure 7.** Quantile–quantile plots comparing the sea levels from the four 1980–1999 CC simulations versus the B-M simulation. Note that for clarity, only the "+" symbols are used to denote the 0.1, 1, 10, 50, 90, 99 and 99.9th percentiles.

From the above analysis we conclude (1) that tide–surge interaction does exist, particularly over the shallow shelf areas in the northwest, northeast and Bass Strait where large tidal amplitudes enhance these interactions. The interactions in these locations affect both the timing and height of the surge. The effect on timing is particularly important for operational forecasting considerations. However, our analysis also shows (2) that there is little overall difference in the magnitudes of the highest weather-driven events (i.e. $\zeta_R$ and $\zeta_M$).

This suggests that for the remainder of this study in which we are dealing with future changes in weather conditions and their effects on sea levels, the omission of tidal forcing in the hydrodynamic simulations forced by climate models is not likely to alter the overall conclusions regarding changes to extreme sea levels (Williams et al., 2016).

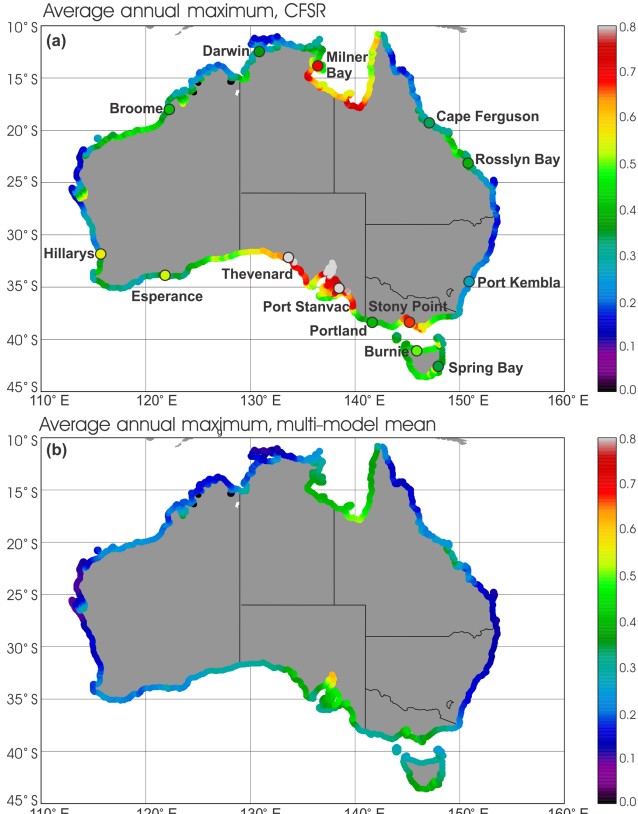

**Figure 8.** The average annual maximum surge over 1980–1999 from the B-M simulation (**a**) and the average annual maximum sea level of the four 1980–1999 CC simulations (**b**). The values derived from tide gauges over the period 1993–2012 are shown by the large circles (**a**). Units are in metres.

## 4 Climate change results

In this section, the primary focus is on changes in ESLs simulated by the climate change experiments listed in Table 2. First, quantile–quantile plots between the current climate (1980–1999) CC simulations and the B-M simulation are created to examine the comparative performance of the different climate models under present climate conditions. Then the differences between the present and future climate conditions are examined.

### 4.1 Comparison with current climate

Figure 7 displays quantile–quantile sea level plots. They are used to compare the performance of the four CC experiments over the current climate period with those from the baseline (B-M) simulation. The figure suggests that the different climate models perform reasonably in modelled sea levels for the lower percentile ranges. The sea level response across the upper percentile range from the climate models over the current climate period is only on par with the baseline experiment for Spring Bay, while Port Kembla, Cape Ferguson

and Portland, Rosslyn Bay, Milner Bay, Broome, Thevenard, Port Stanvac, and Stony Point display lower sea levels. For Darwin the lower percentiles are also overestimated by all models. Of the four simulations, CC-I performs the worst for Broome, Milner Bay, Thevenard and Port Stanvac. CC-H performs the best for Port Stanvac and Thevenard.

The average annual maximum sea levels from the B-M simulation are shown in Fig. 8a together with values from the tide gauge residuals over 1980–1999. From Portland to Broome (anticlockwise), the B-M model is able to represent both magnitude and spatial variation in ESLs well. However, at Hillarys on the west coast and Albany on the southwest coast the model underestimates the extremes. This underestimation may be partly due to the contribution of wind waves to ESLs (i.e. through wave set-up), which is not considered in this study. A second, potentially larger contributor is sea level, variably associated with baroclinic forcing and the Leeuwin Current (McInnes et al., 2016). ESLs were also underestimated in this same region in the study of Haigh et al. (2014a), which, like this study, did not consider wave-driven or baroclinic processes influencing sea level. Model values are also underestimated at Port Stanvac, and this may be due to the poor model resolution of the Gulf of St. Vincent in which Port Stanvac is located.

Figure 8b shows the ensemble-average annual maximum sea levels of the four CC simulations. Results show that the climate model forcing leads to overall lower sea level extremes around the coastline of Australia compared to the baseline (B-M) simulation. This is likely to be at least partially due to the lower spatial and temporal resolution in the CC forcing (Table 1) compared to B-M. However, the variation in the ESL magnitude around the coastline is generally well captured, with higher sea levels in the Gulf of Carpentaria and the southeastern coastline and Tasmania compared to the east and west coast regions.

We note that the skill of eight CMIP5 models in reproducing variables of surface temperature, precipitation and air pressure over continental areas by Watterson et al. (2014), including the four used here, led to model skill rankings which were markedly different to those determined by Hemer and Trenham (2016) in assessing global wind wave climate skill using wind forcing from the same models. This highlights the need to assess the skill of the GCMs according to the task for which they are being used.

### 4.2 Seasonal mean maximum sea level change

To understand how seasonal changes in atmospheric forcing affect both the seasonal or interannual and short-term (storm surge) sea level variations, the average of the largest sea level events per season over each set of 20 seasons is calculated and the 1980–1999 average values are subtracted from those of 2080–2099 (Fig. 9) for each of the CC simulations. The largest positive anomalies of up to 0.1 m are seen in the Gulf of Carpentaria in DJF in the CC-A and CC-H simulations.

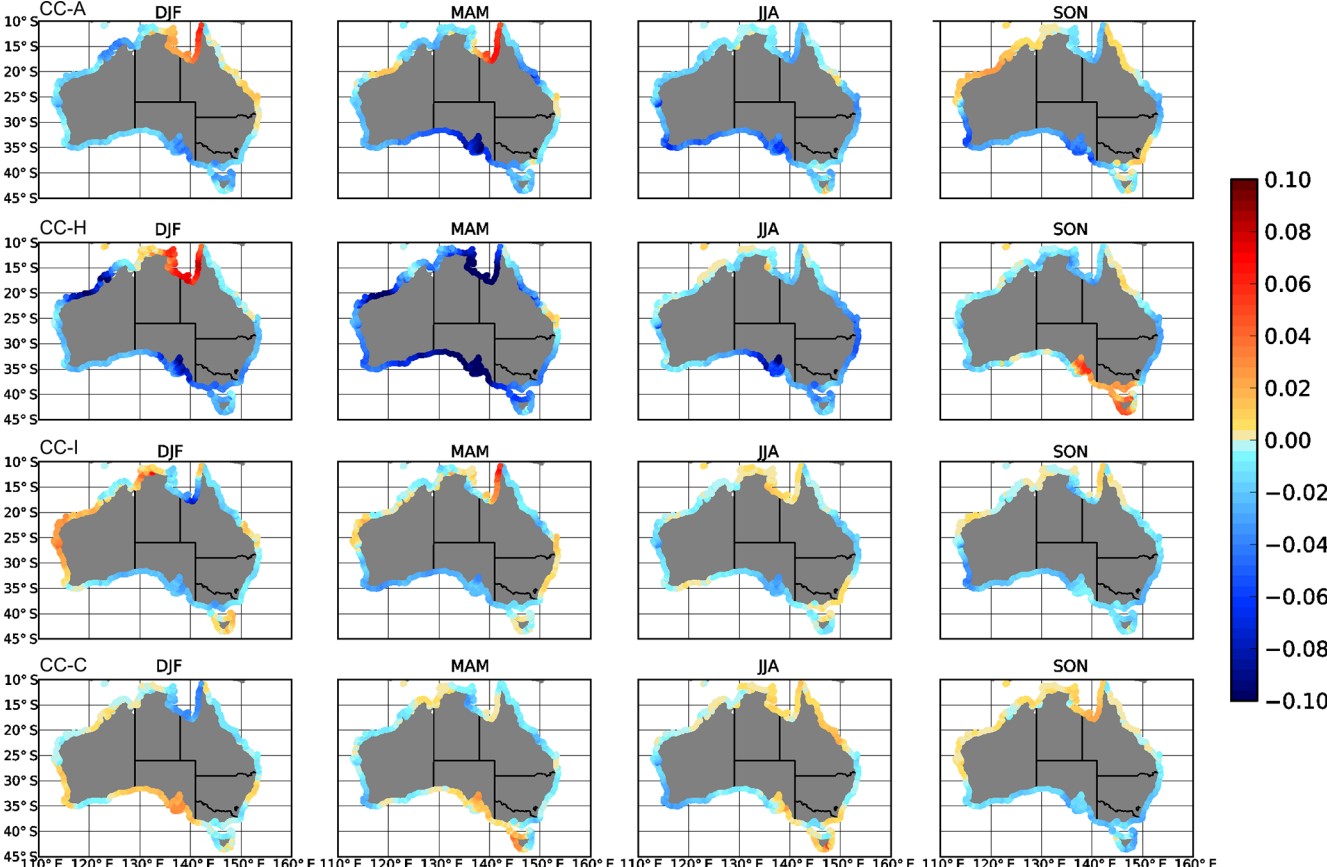

**Figure 9.** Difference in the average of the seasonal maximum sea level between 2081–2100 and 1981–1999 for the different CC models and seasons indicated. Units are in metres.

The positive anomalies extend to MAM in CC-A, are also positive in CC-C but are negative by up to −0.1 m in CC-H. Along the southern mainland coastline, the changes are generally small and mostly negative, consistent with results reported in Colberg and McInnes (2012). However, positive changes are evident in CC-H in SON and CC-I in DJF and MAM over the southeast of the mainland and Tasmania. In the east and west coastal regions, the changes across models are typically small and within the range of ±0.04 m.

To better understand the atmospheric forcing changes responsible for these changes in sea level variability seen in the CC-A simulation between present and future time slices, the change in the seasonal mean and standard deviation (SD) of the wind speed from the ACCESS1.0 is shown in Fig. 10. Also shown in Fig. 10a is the zero contour line of the zonal wind speed from 1980–1999 (blue) and 2080–2099 (red). This contour line identifies the delineation between the monsoon northwesterlies and trade wind easterlies over northern Australia during DJF and the subtropical ridge separating trade easterlies from midlatitude westerlies over southern Australia throughout the year.

During DJF the eastward shift in the zero contour of the zonal wind in 2080–2099 DJF is accompanied by a general increase in wind speed across tropical Australia and wind SD within the Gulf of Carpentaria. This suggests there is a greater influence of northeast monsoon winds on the Gulf of Carpentaria, which provide favourable conditions for increased sea levels in the gulf (Oliver and Thompson, 2008). The CC-H simulations produce a similar increase in sea levels in the gulf during DJF, also related to northwest monsoon winds penetrating further east and increased variability in this region. The reasons for the positive anomalies in the ACCESS1.0 and the CC-C simulations in MAM are less clear since both simulations show a decrease in mean winds and variability in the Gulf of Carpentaria (not shown).

Along the southern coastline of the continent and Tasmania, there is a tendency for a decrease in ESLs in most seasons of the models. As illustrated in Fig. 10 for CC-A, this is related to the southward movement of the subtropical ridge, reduced wind variability and the greater frequency of non-storm-surge-producing easterly winds. In CC-H in SON, positive anomalies in sea level are seen, and this is related to both an increase in westerlies over Tasmania and a strong increase in SD (not shown). The weak increase in CC-I in DJF is related to the minimal southward movement of the midlat-

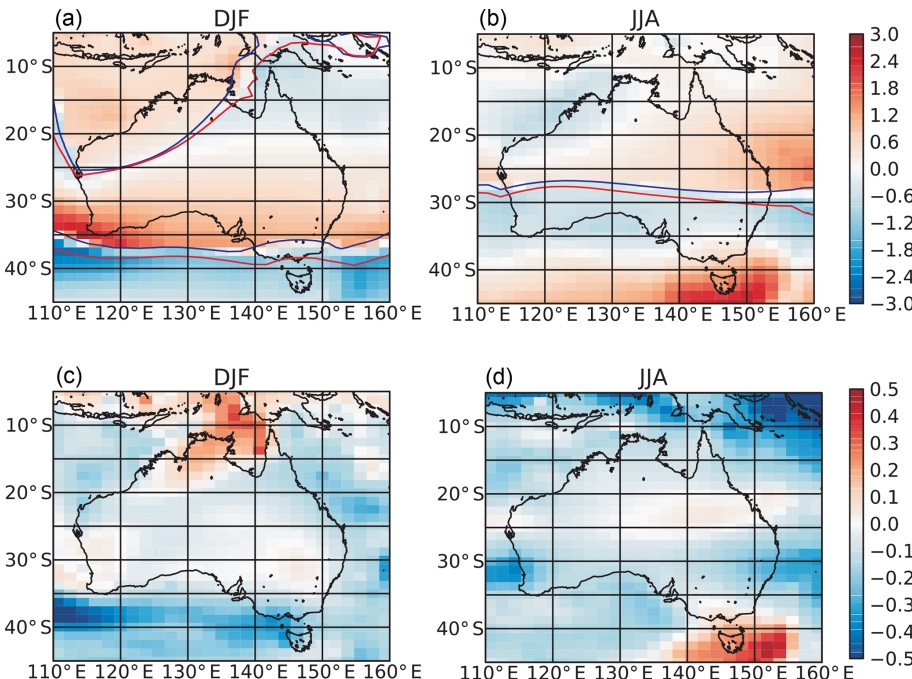

**Figure 10. (a, b)** CC-A modelled changes in wind speed for DJF and JJA with the zero of zonal wind speed shown as a contour in blue for current climate and red for future climate. **(c, d)** Standard deviation of wind speed for DJF and JJA as modelled by CC-A.

itude storm belt together with an increase in the SD in that model.

The overall projected changes to maximum ESL events around Australia are summarised in Fig. 11. These ensemble differences are generated by finding the difference between the maximum sea level for the 1990–1999 and 2080–2099 time periods for each of the CC ensembles members. Since each time period is 20 years, this equates to the (empirical) change in the 1-in-20-year average recurrence interval; the minimum, average and maximum of these ensemble differences are shown in Fig. 11a, b and c and give an indication of uncertainty. Additionally, the values of ESL are hatched where the model solutions differ in sign, indicating inter-model variability. The minimum changes are negative around the entire coastline indicating an average decrease in the approximate 20-year average recurrence interval in the range of 0 to 0.2 m. The largest projected decreases are on the northwestern shelf and the central west and south coasts. The average change across the four models is weakly negative around most of the coastline with weak positive anomalies evident along parts of the north, the Gulf of Carpentaria and southern Tasmania. The ensemble maximum changes show weak positive anomalies of up to 0.04 m along the southeast and east coast. The largest positive changes of up to 0.15 m occur on the eastern side of the Gulf of Carpentaria, the central north coast and parts of the northwest and west coast. Negative anomalies occur on the central south and southwest coasts. Overall, model results are fairly robust over the southern coastline where all models suggest a decline in maxi-

mum sea levels. Large areas particularly over the north exist where changes in maximum ESL could go either way depending on the atmospheric model used. This may indicate possible uncertainties in parameterising atmospheric convection in climate models over the tropics, which in turn strongly influences monsoonal winds and sea level set-up in the Gulf of Carpentaria. It is worth noting that Vousdoukas et al. (2018) project changes for the Australian coastline in a six-member ensemble containing one model in common with the present study (ACCESS1.0) and, for 2100 under RCP 8.5, find largely uncertain changes in the Gulf of Carpentaria. There are mostly negative changes around the eastern, southern and western coastlines, positive changes across Tasmania and southeastern Australia, and uncertain changes along the southwestern mainland coastline and the Gulf of St Vincent which are somewhat consistent with the findings in this study.

## 5   Summary and concluding discussion

In order to investigate characteristics of ESLs, a depth-averaged hydrodynamic model covering Australia was implemented at 5 km spatial resolution and baseline simulations carried out over the period 1981 to 2012 with hourly atmospheric and tidal forcing. Overall, simulations of longer-term (seasonal and interannual) and short-term (weather-driven) variations in sea level compare well with those measured at tide gauges, with differences largely reflecting the absence

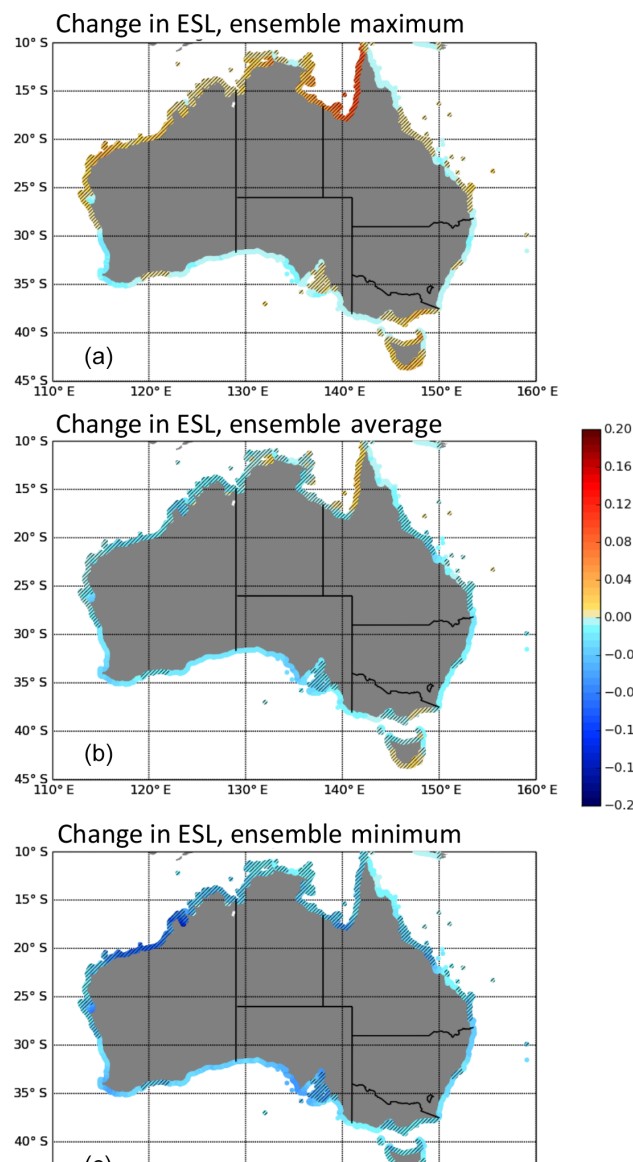

**Figure 11.** Projected change in the 1-in-20-year average return interval (ARI) extreme sea level (residuals) for the model ensemble. **(a)** Ensemble minimum. **(b)** Ensemble average. **(c)** Ensemble maximum. Hatched areas indicate where models disagree on sign. Units are in metres.

of baroclinic forcing in the model. The modelled tides agree well with observations everywhere except the Gulf of Carpentaria, where the O1 and K1 constituents were underestimated by the model, and the southwestern coast, where the M2 and S2 constituents were underestimated. The effect of tide–surge interaction on the amplitude of the meteorological component of sea level extremes (e.g. storm surge) was found to be small for much of the coastline, the main effect of the interaction being on the timing of the peak sea levels rather than the annual maximum surges or residuals.

This suggested that in climate-model-forced hydrodynamic simulations that assess how atmospheric circulation changes affect ESLs, tidal forcing could be neglected. This is further supported by the finding (across a large number of north Atlantic tide gauges) that while tide–surge interaction may affect the timing of maximum water levels, tides have no direct effect on the magnitude of storm surge (Williams et al., 2016).

Hydrodynamic simulations were carried out over the periods 1980–1999 and 2080–2099 using forcing from four CMIP5 climate models run with the RCP 8.5 emission scenario. Changes in ESLs were generally small and mostly negative along much of the coastline. However, in some areas ESL changes were sensitive to the movement of major atmospheric circulation patterns. This was because of factors such as bathymetric depths and coastline orientation in relation to the weather forcing that favoured the occurrence of certain sea level extremes. For example, the Gulf of Carpentaria exhibited relatively large increases in ESLs in the climate models that simulated eastward movement of the northwest monsoon during the DJF season. However, since only two of the four climate model simulations simulated this change in the future climate, the finding is uncertain. Along the mainland south coast, there was a greater tendency for the models to indicate a reduction in ESLs in the future, particularly during winter, which is also consistent with the finding of Colberg and McInnes (2012) using CMIP3 and regional climate models for the atmospheric forcing and somewhat similar to the study of Vousdoukas et al. (2018) using an ensemble of six CMIP5 GCMs for the atmospheric forcing.

With regards to the projected ESL changes, we note several important caveats. First, the changes are subject to large uncertainty due to the small number of CMIP5 models used to force the hydrodynamic model. Furthermore, certain important drivers of ESLs may be poorly represented in climate models in general, e.g. tropical cyclones (TCs). Previous studies (e.g. Haigh et al., 2014b) have demonstrated that numerical hindcasts (or projections) of several decades typically provide poor estimations of non-tidal ESLs over tropical Australia. This is due both to a lack of sufficient resolution in most available atmospheric models and the low-frequency and high-impact nature of locally landfalling TCs, which generally provides low statistical confidence in related ESLs at any given location. Modelling a large numbers of synthetic cyclones (e.g. Haigh et al., 2014b; McInnes et al., 2014) can address this shortcoming to some extent although a question in terms of statistical robustness may remain. Furthermore, areas which experience high interannual and decadal sea level variability may also require specialised statistical treatment and or very long model runs (i.e. greater than several decades) to accurately characterise non-tidal ESLs. This is illustrated in the results presented here: many areas shown to have a high dependence on ENSO variability (e.g. Fig. 4 in McInnes et al., 2016b) coincide with areas where the projected ESL changes disagree in sign

in this study (Fig. 11). Thus, results presented here should be seen in the context of the limitations of the available data and/or downscaling techniques used. We demonstrate a robust change in ESLs over southern Australia, while future changes over tropical Australia remain largely uncertain due to the spatial resolution of CMIP5 climate models and the use of computationally tractable time slices (here 20 years). Future studies may address these uncertainties by better exploring the uncertainty space, e.g. by considering a larger ensemble of hydrodynamic simulations forced with higher-resolution climate models that better capture important small-scale meteorological features or by perturbing characteristics of historical storms to produce plausible future synthetic storm libraries (McInnes et al., 2014). We also note that wind waves contribute to sea level extremes, and these effects and their potential changes need to be assessed for a more complete understanding of the changes to sea level extremes (e.g. Hoeke et al., 2015). The increasing availability of wave climate change assessments (e.g. Hemer et al., 2013; Hemer and Trenham, 2016) will facilitate future efforts in this regard. Also, while previous studies similar to this one have focused on changes to ESLs and coastal inundation (e.g. Colberg and McInnes, 2012; McInnes et al., 2013), consideration of changes to other variables, including currents, is emerging (e.g. Lowe et al., 2009). Changes to wind-driven coastal currents, which could be considered using the modelling framework presented in this study (but are beyond the scope of this paper), are also potentially important in the context of coastal erosion and shoreline change (Gornitz, 1991; O'Grady et al., 2015).

*Data availability.* [TS1] https://doi.org/10.4225/08/5a7280a3a0d2a.

*Author contributions.* FC and KLM jointly conceived the study. FC configured the model, prepared input data sets, calibrated and ran all model simulations and contributed to the analysis of data and the writing of the paper. KLM, JOG and RH contributed to the analysis of results and the writing of the paper.

*Competing interests.* The authors declare that they have no conflict of interest.

*Acknowledgements.* The Australian Climate Change Science Program and the Earth System and Climate Change Hub of Australian Government's National Environmental Science Program are acknowledged for funding this research. We also acknowledge the CSIRO High Performance Computing facility, which provided computational support for the work described in this study.

Edited by: Joaquim G. Pinto
Reviewed by: two anonymous referees

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
