# Peer review of "Atmospheric Circulation Changes and their Impact on Extreme Sea Levels around Australia"

_Natural Hazards and Earth System Sciences, 2018_

## Referee Comment (RC1) · Anonymous Referee #1 · 5 Aug 2018

This paper investigates four projections of sea levels for the Australian coast and associates them to changes of atmospheric circulation. It first validates the wave model used. Subsequently, it compares a present (1981-1999) and a future (2081-2099 with RCP8.5 forcing)time slices to identify a climate change signal. The study is not methodologically new, but it aims at providing new information for the Australian coast. My concern is that this information is apparently not very conclusive, meaning that, according to my interpretation of results, disagreement among model prevents reaching robust conclusions except for the decrease of extremes along a relatively small part of the Australian Southern coast.

The duration (20 years) of the analyzed time periods is lower than 30 year minimum duration often recommended. Effects of multidecadal variability could in this case hide a

climate change signal that is not sufficiently strong. I think that it should be investigated whether using longer time slices could have produced more robust results.

I think (see my comments below) that this manuscript and the figures should be improved for becoming publishable.

More specific comments:

—-Abstract

1) The abstract should state more clearly the main conclusions. I think the present one is not satisfactory on this respect. The text at lines 15-16 is too generic and not informative on atmospheric circulation changes, while they, being mentioned in the title, should be a main focus of the manuscript. The last lines mention a large increase in extreme sea level during austral summer in the Gulf of Carpentaria (note that it is difficult to locate it for those not familiar with Australian geography and it is not mentioned in the map of figure 1). However, this conclusion is rather uncertain because only 2 of the 4 models used show such increase (fig.9). Further, the abstract mentions a small reduction of sea level extremes along the southern coast. However, the four models (fig.11) agree only on a relatively small central fraction of the southern coast and on its westernmost tip. A limitation of this study is, in my view, that it is unable to identify significant change in surge extremes (there is very little agreement among models).

—- Introduction

2) at page 1, line 24 it is not clear what authors mean here. Do they mean that storm surges are superimposed to low frequency modulation of sea level (forced by large scale patterns) or that the synoptic forcing of extremes is, in turn, modulated by large scale circulation patterns?

3) it seems to me that the author do not summarize adequately the existing literature. Page1, line 33 to page 2, line16, This paragraph looks rather incomplete to me. Only

the last four lines refer to Australia. Is it reasonably complete list of available studies for Australia ? After a quick search with google scholar have found also

- McInnes, K.L., Macadam, I., Hubbert, G.D. et al. Nat Hazards (2009) 51: 115.

- Church JA, Hunter JR, McInnes KL, White NJ (2006). Aust Meteorol Mag 55:253–260

Are they not relevant?

The rest is for European Seas and it looks a very incomplete reference to a very rich literature, with many studies published for the North and the Mediterranean Seas. Again, just searching with scholar, I have found

Vousdoukas, M.I., Voukouvalas, E., Annunziato, A. et al. Clim Dyn (2016) 47: 3171. https://doi.org/10.1007/s00382-016-3019-5

Woth, K., Weisse, R. & von Storch, H. Ocean Dynamics (2006) 56: 3. https://doi.org/10.1007/s10236-005-0024-3

Conte D.and LionelloP. (2013) https://doi.org/10.1016/j.gloplacha.2013.09.006

Androulidakis YS et al.(2015) https://doi.org/10.1016/j.dynatmoce.2015.06.001

Lionello P.et al (2017) https://doi.org/10.1016/j.gloplacha.2016.06.012

Debernard J, Røed L (2008) Tellus 60:427–438. doi:10.1111/j.1600–0870.2008.00312.x

R.Weisse et al (2012) https://doi.org/10.1016/j.ocecoaman.2011.09.005

. . . and this list does not mean to be complete

The fact that the authors do not adequately summarize the existing literature applies also to the following paragraphs on interactions between storm surge and sea level rise and on tropical cyclones

4) Page 3 lines 12 to 19.Should be better explained what is new in this study. Which

new information is missing and authors aim at providing?

——In Model description and methods

5) Section 2.1. which fraction of the total tidal amplitude is explained by using only 8 components?

6) Section2.3 the problem with introduction of tides in model adopting 360 day long year does not appear relevant because tides are not included in the climate change experiments (authors write this a few lines below)

7) Page 5 , lines 11-13, RMSE, STDE and correlation are weak metrics for validation of extremes in a time-series. High correlation and low RMSE can be obtained also if extremes are poorly reproduced. Further, to validate a model percent errors should be considered, particularly for extremes. To compare magnitude of the error to the magnitude of the observed value is important.

8) Page 5, lines 4.It is not clear to me how is the seasonal variability component defined and computed in this study

9) Page 5, line 31. If the 30-day running mean is subtracted to the signal, I expect that the steric contribution on the residual is small

——Sea level residuals.

10) It is not clear to me what we learn from the considered examples. What have been the criteria for their selection

11) Page 6, lines 33-34 to blame the inaccurate meteorological forcing is often correct, but it is also an easy way out. Can the authors provide an argument to support this?

12) Page 7, line 13. It is not clear why in this specific location wave set up is expected to be a relevant contribution and could explain the underestimated sea level by the model. This should be explained in terms of location of the gauge and morphology of the sea bottom (including depth).

–––Tide-surge interactions

13) Page 8 lines 6-11. To which figures do these sentences refer?

14) Authors consider the total sea level ZTM , Its tidal ZT and meteorological ZM components, all computed separately by independent simulation. Defining the residual ZR= ZTM - ZT, they find that peaks (ranks) of ZR and ZM agree and conclude that time-surge interaction is negligible. However, this is in contrast with the lack of agreement between ZTM and ZT + ZM , which shows that tides substantially decrease the importance of the meteorological contribution to sea level extremes. Therefore, to me it seems that tides are not relevant for computing correctly the maxima of the storm surge, but actual sea level maxima are affected (decreased) by tide-surge interaction (practically high tidal levels reduce the contribution of the surge to the maxima). Further, the whole analysis applies at the location of the tide gauge. I suspect that at the actual coastal line, at the shore, analysis can produce different results.

––– comparison with current climate.

15) The statement that "climate models overall perform well" is too positive, considering the tendency of all simulations to underestimate high quantiles is some locations (fig.7).Such underestimate is particularly large for inmcm (note that the annotation in the figure is not consistent with the text which refers to this simulation as CC-I). Model simulations substantially underestimate extremes at several locations.

–––Seasonal mean maximum sea level change

16) I find this part should be improved in several aspects 16.1) It discusses the multimodel mean at annual scale, and only individual models at seasonal scale. 16.2) There is no indications whether changes are statistically significant for individual models. I suggest to mask in figure 11 (central panel) values when models do not agree on the sign of the change or add, anyway, an indication of the level of consensus among models. 16.3) There is a discussion of the link of the observed changes of extremes

with changes of wind speed. However, it is not clear why changes of mean speed are relevant for extremes and whether figure 10 is a multimodel mean or it represents the winds driving the CC-A simulations.

—–Some minor comments

The authors are native English speakers, while I am not. However, I find that the text in some could be improved. Examples in the abstract

Line 7 "short term": Do authors mean at the monthly, annual or decadal scale? I think they mean "high frequency" here

Line 7 attendant->expected

Line 10 conditions -> observations

Line 11 delete simulation

—-Quick comments on the figures

Figure 1: station names are too small

Figure 3 , titles and axis labels not readable

Figure 4, panel should have a larger size and blank areas among them should be reduced. The lowest quantile is 0.1 and , consistently the largest is 99.9 according to the caption. However, there are blue points above the highest red point (denoting the 99.9 quantile) which high quantile are shown here?

Figure 5 arrows (wind speed) are not visible (they are too small). I suggest to add dots to mark the position of the station considered in each panel

Figure 6 caption does not describe it properly

. . .in general I think captions could be improved and describe should more exhaustively the content of figures

---

## Referee Comment (RC2) · Anonymous Referee #2 · 4 Sep 2018

The paper by Colberg et al. investigates the performance of a medium resolution hydrodynamic model to simulate observed extreme sea levels (ESLs) for the Australian coastline and to estimate potential changes in ESLs as response to a future climate. Considering all individual water level components is computationally demanding especially with regards to scenario runs and this is why the authors conducted sensitivity analyses to estimate the effect of tides on total water levels. Their conclusion is that tide-surge interaction is strong (at least at some parts alomg the coast) for individual extremes but may be neglected for statistics over longer periods. For me, this is not very conclusive as potential changes in tides are also not taken into account. I understand somehow that changes in the met. forcing alone may be visible in a surge-only run and potential changes may (roughly) be inferred thereof but only if relative changes in

the met. iduced component are investigated. Climate change will also affect the base water level (MSL) which has not been considered in your experiments, right? From SLR, the propagation of the surge will be affected influencing the timing (and hights) of surge events. Furthermore, also the tidal propagation may/will change with SLR having the potential to further increase water levels and partly compensate for the "mostly" negative trend in ESL changes you reported stemming from the met. only approach.

The period of 20 yrs you consider for the future climate conditions are too short to draw robust conclusions. Usually a period of 30yrs is used to estimate changes in the met. forcing. Please consieder extending your modelling or discuss why you chose this short period, how it affects your results.

The paper would benefit from improvements on the above mentioned aspects. Further specific comments are below:

page 3, line 30: 1' x 1', x missing

page 7, line 26: Due to computational constraints, we demonstrate that... From my point of view, this is not a good argumentation

page 9, line 8: BM covers 1981-2012, right? so the common period is '81-'99. Should be clear

page 9, line 18: Albany not shown in Figure 1

Fig1: Could be helpful to show the average tidal range (e.g. based on TPXO) over the entire area

Fig.2: Please define the dots (semi- and diurnal)

Fig.4: All R2s show values of ∼1. This is a bit misleading, as most stations over- and/or underestimate the extremes. Also the R2 is not mentioned

Fig. 6: Units missing; please highlight meaning of surge and residual again; for me, the figure shows a clear tide-surge interaction which cannot be neglected. Also for the

largest events as e.g. in Rosslyn Bay or Darwin

Fig. 8: Portland not given in the Fig. , what is happening at the northern part (Milner Bay)

All figures would benefit from detailed captions.
* * *

---

## Author Comment (AC1) · 16 Oct 2018

[10pt]article color hyperref  In the text below, the reviewer's comments are shown in black, while our (the authors') responses are shown in red text.
* * *
The duration (20 years) of the analed time periods is lower than 30 year minimum duration often recommended. Effects of multidecadal variability could in this case hide a climate change signal that is not sufficiently strong. I think that is should be investigated wheather using longer time slices could have produced more robuts results.

[Figure]

The choice of twenty-year time slices was to align the hydrodynamic model output to wave model simulations carried out using the same climate models over the same time period that was published in Hemer and Trenham (2016). Our aim was to be able to couple hydrodynamic extremes with wave-induced extremes (e.g. wave setup or runup) in future work. We acknowledge that 20 years may be too short to assess the role of future changes to interannual variability (i.e. ENSO) on weather events that cause extreme sea levels such as tropical cyclones, but as we already note, the GCMs do not adequately resolve TCs anyway so the focus of our study is on the contribution of large scale circulation changes to extreme sea levels. We feel that 20-year time slices are adequate for assessing how large scale circulation changes will affect drivers of sea levels around much of Australia's coast where seasonally varying weather systems are a major cause of extreme sea levels.

(Hemer, M. A. and C. E. Trenham, 2016: Evaluation of a CMIP5 derived dynamical global wind wave climate model ensemble. Ocean Modelling, 103, 190-203, "doi:https://doi.org/10.1016/j.ocemod.2015.10.009".)

Abstract

The abstract should state more clearly the main conclusions. I think the present one is not satisfactory on this respect. The text at lines 15-16 is too generic and not informative on atmospheric circulation changes, while they, being mentioned in the title, should be a main focus of the manuscript. The last lines mention a large increase in extreme sea level during austral summer in the Gulf of Carpentaria (note that it is difficult to locate it for those not familiar with Australian geography and it is not mentioned in the map of figure 1). However, this conclusion is rather uncertain because only 2 of the 4 models used show such increase (fig.9). Further, the abstract mentions a small reduction of sea level extremes along the southern coast. However, the four models (fig.11) agree only on a relatively small central fraction of the southern coast and on

its westernmost tip. A limitation of this study is, in my view, that it is unable to identify significant change in surge extremes (there is very little agreement among models).

We removed the sentence in question. We understand that there is a limited amount of agreement between the different modle simulations and changed our wording around it, we give more detailed information regarding the SSH changes in the abstract. Overall we argue that the disagreement in responses between the differently forced model simulations and the difference in seasonally in their response is in itselft is a valuable result. It may emphasize that we need to work towards a better understanding of parametrized physics in climate models. These unresolved phyics may very well drive a large amount of uncertainty and may lead to large intra-model differences. We have changed the manuscript as to put a stronger emphasize on this aspect which we did not do so before.

We marked the location of the Gulf of Carpenteria in Figure 1 !

Introduction

page 1, line 24 it is not clear what authors mean here. Do they mean that storm surges are superimposed to low frequency modulation of sea level (forced by large scale patterns) or that the synoptic forcing of extremes is, in turn, modulated by large scale circulation patterns?

OK - We reworded this part of the manuscript and hopefully made this section more clear.

It seems to me that the author do not summarize adequately the existing literature. Page1, line 33 to page 2, line16, This paragraph looks rather incomplete to me. Only the last four lines refer to Australia. Is it reasonably complete list of available studies for Australia ? After a quick search with google scholar have found also

Yes, we agree with the reviewer and have added additional relevant references

Page 3 lines 12 to 19.Should be better explained what is new in this study. Which new information is missing and authors aim at providing?

OK - we strengthened the introduction to emphaisze what is new in this modelling study. SSH changes driven by synoptic weather changes for the whole australian coastline has not ben investigated before.

Section 2.1. which fraction of the total tidal amplitude is explained by using only 8 components?

The dominant tidal constituents are the diurnal constituents, K1, O1, P1, Q1, and S1, and the semidiurnal constituents M2, S2, N2, and S2 (Wollanksi, and Elliot, 2016). Other constituents that typically may contribute non-trivially to overall coastal tidal amplitudes include higher-frequency non-linear shallow water "overtides" and annual and semiannual constituents (which are typically due more to seasonal oceanographic and meteorological variability, rather than astronomical forcing). The ROMS model is capable of dynamically reproducing both of these types constituents (at least to some degree). We therefore follow a frequent convention used for shelf-scale models and the 8 major tidal constituents at the model boundaries.. It is not necessarily possible (or desirable) to obtain higher order tidal constituents via global tidal models.

Section2.3 the problem with introduction of tides in model adopting 360 day long year does not appear relevant because tides are not included in the climate change experiments (authors write this a few lines below)

Yes, we agree with the reviewer and removed this paragraph from manuscript.

Page 5 , lines 11-13, RMSE, STDE and correlation are weak metrics for validation of extremes in a time-series. High correlation and low RMSE can be obtained also if extremes are poorly reproduced. Further, to validate a model percent errors should be considered, particularly for extremes. To compare magnitude of the error to the magnitude of the observed value is important.

We are aiming to compare the analyses to that given by Haigh et al, 2014a. The aim thus was to show that the model captures atmospheric driven variability generally well. We asses extremes via qq plots in Figure 4.

8 Page 5, lines 4.It is not clear to me how is the seasonal variability component defined and computed in this study

We follow the methodology of Haigh et al,2014a. The seaosnal component is calculated by using a 30day running mean over the detided and detrended time series. This removes basically the high frequency (weather driven) variability. We changed the paragraph to make this more clear.

Page 5, line 31. If the 30-day running mean is subtracted to the signal, I expect that the steric contribution on the residual is small

We are not quite sure what the reviewer means. We use the 30 day running mean to tease out the seasonal signal. This is also done in accordance to Haigh et al, 2014a.

Sea level residuals. It is not clear to me what we learn from the considered examples. What have been the criteria for their selection

The examples have been selected to illustrate the main weather systems that cause storm surges along different coastal regions in Australia. The year 1997 was selected as it contained examples of extreme sea levels along each coastal region examined.

We are adding additional references in the paper.

Page 6, lines 33-34 to blame the inaccurate meteorological forcing is often correct, but it is also an easy way out. Can the authors provide an argument to support this?

Yes, this is true. Arguably there are acouple of reasons why the model is not able to reproduce SSH anomalies correctly (1) representation error of model and atmospheric (i.e. grid resolution, bathymetry errors, coastlines not resolved properly, errors in the atmospheric forcing, limited temporal resolution), (2) model physics - a 2D model will only ever resolve the first barotropic mode of coastally trapped waves. Higher order modes may be necessary to properly account for all the variability. These points are of course generic and speculative.

Page 7, line 13. It is not clear why in this specific location wave set up is expected to be a relevant contribution and could explain the underestimated sea level by the model. This should be explained in terms of location of the gauge and morphology of the sea bottom (including depth).

We agree with the reviewer and changed the paragraph. In fact what appears to happen here is explained by the following: The missing peak can be explained by a not propoerly cpatured/ modelled coastlly trapped wave (CTW). Studies like Woodham et al, 2013 suggest speeds of CTW between 2-4m/s. CTW travel antiklockwise around Australia. It takes about 5-6days for CTW to travel the distant from port kembla to Rosslyn Bay. On the 10th may a coastal low produced a surge in Port Kembla that excuted CTW. According to Woodham et al, CTW can cause sea level elevations of 0.25m which is in the order of what has been observed at Rosslyn Bay about 5-7 days later. The ROMS model does not capture this elevation,. This may potentially be due to its barotropic nature which does not allow higher order (bariclinic) modes of CTW to develop. Unresolved bathymetric features over the Great Barrier Reef are

also candiates for explaining the model behaviour.

Page 8 lines 6-11. To which figures do these sentences refer?

These sentences refer to Figure 6. We made appropriate changes in the text.

Authors consider the total sea level ZTM , Its tidal ZT and meteorological ZM components, all computed separately by independent simulation. Defining the residual ZR= ZTM - ZT, they find that peaks (ranks) of ZR and ZM agree and conclude that time-surge interaction is negligible. However, this is in contrast with the lack of agreement between ZTM and ZT + ZM , which shows that tides substantially decrease the importance of the meteorological contribution to sea level extremes. Therefore, to me it seems that tides are not relevant for computing correctly the maxima of the storm surge, but actual sea level maxima are affected (decreased) by tide-surge interaction (practically high tidal levels reduce the contribution of the surge to the maxima). Further, the whole analysis applies at the location of the tide gauge. I suspect that at the actual coastal line, at the shore, analysis can produce different results.

We agree with the assertion by the reviewer which is what we wrote in the manuscript. We were reviewing the paragraph to make this more clear.

The statement that "climate models overall perform well" is too positive, considering the tendency of all simulations to underestimate high quantiles is some locations (fig.7).Such underestimate is particularly large for inmcm (note that the annotation in the figure is not consistent with the text which refers to this simulation as CC-I). Model simulations substantially underestimate extremes at several locations.

We understand the the formulation used here may sound too positive and we changed the paragraph to accomodate the reviewers concern. However we would also like

to point out that the second part of the sentence in question puts the first part in perspective: "...climate models overall perform well in terms of generating a sea level response in the ocean model for the lower percentile ranges". So we are saying what the reviewer questions in his comment. Furthermore, we go on to say that the sea level response for upper percentile ranges is on par for certain stations and not on par for other stations.

Seasonal mean maximum sea level change I find this part should be improved in several aspects 16.1) It discusses the multimodel mean at annual scale, and only individual models at seasonal scale.

The shown multimodel mean for the annual average is to compare results from CFSR forced surge model with observations and to show that the CMIP5 forced models show similar results in terms of overall distribution.

There is no indications whether changes are statistically significant for individual models. I suggest to mask in figure 11 (central panel) values when models do not agree on the sign of the change or add, anyway, an indication of the level of consensus among models.

OK – we understand the reviewers concern and added his suggestions to the image.

There is a discussion of the link of the observed changes of extremes with changes of wind speed. However, it is not clear why changes of mean speed are relevant for extremes and whether figure 10 is a multimodel mean or it represents the winds driving the CC-A simulations.

Yes, we agree with the reviewer and clarified this. Figure 10 demonstrates a possible mechanism that may explain the observed increase in extremes seen for ACCESS-R

and HadGEM foreced model simulations. Stronger mean monsoonal winds will pile up more water over the GOC. This in turn will increase the likelihood of extremes to happen.

---

## Author Response (AR1)

Please find our response to reviewers below. Our responses to the reviewers comments are in red text.

**Reviewer 1**
The duration (20 years) of the analed time periods is lower than 30 year minimum duration often recommended. Effects of multidecadal variability could in this case hide a climate change signal that is not sufficiently strong. I think that is should be investigated wheather using longer time slices could have produced more robuts results.
The choice of twenty-year time slices was to align the hydrodynamic model output to wave model simulations carried out using the same climate models over the same time period that was published in Hemer and Trenham (2016). Our aim was to be able to couple hydrodynamic extremes with wave-induced extremes (e.g. wave setup or runup) in future work. We acknowledge that 20 years may be too short to assess the role of future changes to interannual variability (i.e. ENSO) on weather events that cause extreme sea levels such as tropical cyclones, but as we already note, the GCMs do not adequately resolve TCs anyway so the focus of our study is on the contribution of large scale circulation changes to extreme sea levels. We feel that 20-year time slices are adequate for assessing how large scale circulation changes will affect drivers of sea levels around much of Australia's coast where seasonally varying weather systems are a major cause of extreme sea levels.

(Hemer, M. A. and C. E. Trenham, 2016: Evaluation of a CMIP5 derived dynamical global wind wave climate model ensemble. Ocean Modelling, 103, 190-203, doi:https://doi.org/10.1016/j.ocemod.2015.10.009.)

We have added a brief discussion on this at the end of paragraph 1 of section 2.3

*Abstract*
The abstract should state more clearly the main conclusions. I think the present one is not satisfactory on this respect. The text at lines 15-16 is too generic and not informative on atmospheric circulation changes, while they, being mentioned in the title, should be a main focus of the manuscript. The last lines mention a large increase in extreme sea level during austral summer in the Gulf of Carpentaria (note that it is difficult to locate it for those not familiar with Australian geography and it is not mentioned in the map of figure 1). However, this conclusion is rather uncertain because only 2 of the 4 models used show such increase (fig.9). Further, the abstract mentions a small reduction of sea level extremes along the southern coast. However, the four models (fig.11) agree only on a relatively small central fraction of the southern coast and on its westernmost tip. A limitation of this study is, in my view, that it is unable to identify significant change in surge extremes (there is very little agreement among models).
We removed the sentence in question. We understand that there is a limited amount of agreement between the different modle simulations and changed our wording around it, we give more detailed information regarding the SSH changes in the abstract.
Overall we argue that the disagreement in responses between the differently forced model simulations and the difference in seasonality in their response is in itselft is a valuable result. It may emphasize that we need to work towards a better understanding of parametrized physics in climate models. These unresolved phyics may very well drive a large amount of uncertainty and may lead to large intra-model differences. We have changed the manuscript to put a stronger emphasize on this aspect.

The Gulf of Carpenteria (GoC) is now indicated in Figure 1.

*Introduction*

page 1, line 24 it is not clear what authors mean here. Do they mean that storm surges are superimposed to low frequency modulation of sea level (forced by large scale patterns) or that the synoptic forcing of extremes is, in turn, modulated by large scale circulation patterns?
We have reworded this part of the manuscript to clarify our meaning here.

It seems to me that the author do not summarize adequately the existing literature.
Page1, line 33 to page 2, line16, This paragraph looks rather incomplete to me. Only the last four lines refer to Australia. Is it reasonably complete list of available studies for Australia ? After a quick search with google scholar have found also

3) it seems to me that the author do not summarize adequately the existing literature. Page1, line 33 to page 2, line16, This paragraph looks rather incomplete to me. Only the last four lines refer to Australia. Is it reasonably complete list of available studies for Australia ? After a quick search with google scholar have found also –
McInnes, K.L., Macadam, I., Hubbert, G.D. et al. Nat Hazards (2009) 51: 115. –
Church JA, Hunter JR, McInnes KL, White NJ (2006). Aust Meteorol Mag 55:253–260 Are they not relevant?
The rest is for European Seas and it looks a very incomplete reference to a very rich literature, with many studies published for the North and the Mediterranean Seas. Again, just searching with scholar, I have found
Vousdoukas, M.I., Voukouvalas, E., Annunziato, A. et al. Clim Dyn (2016) 47: 3171. https://doi.org/10.1007/s00382-016-3019-5
Woth, K., Weisse, R. & von Storch, H. Ocean Dynamics (2006) 56: 3. https://doi.org/10.1007/s10236-005-0024-3
Conte D.and LionelloP. (2013) https://doi.org/10.1016/j.gloplacha.2013.09.006 Androulidakis YS et al.(2015) https://doi.org/10.1016/j.dynatmoce.2015.06.001
Lionello P.et al (2017) https://doi.org/10.1016/j.gloplacha.2016.06.012
Debernard J, Røed L (2008) Tellus 60:427–438. doi:10.1111/j.1600– 0870.2008.00312.x
R.Weisse et al (2012) https://doi.org/10.1016/j.ocecoaman.2011.09.005 . . . and this list does not mean to be complete
The fact that the authors do not adequately summarize the existing literature applies also to the following paragraphs on interactions between storm surge and sea level rise and on tropical cyclones
We have updated and extended our literature review with additional relevant references

Page 3 lines 12 to 19.Should be better explained what is new in this study. Which new information is missing and authors aim at providing?
We have strengthened the introduction (paragraph 1 and 7) to emphasize what is new in this modelling study. SSH changes driven by synoptic weather changes for the whole australian coastline have not been investigated before.

Section 2.1. which fraction of the total tidal amplitude is explained by using only 8 components?
We follow the common convention used for shelf-scale models, which is to apply the 8 major tidal constituents at the deep water model boundaries. These major constituents are obtained from global tide models. It is not necessarily possible to obtain higher order tidal constituents from global tidal models due to their low coastal resolution, nor is it necessary to do anyway since the tidal heights are applied as a deep water boundary condition where overtides would not occur anyway. The dominant tidal constituents are the semidiurnal constituents $M_2$, $S_2$, $N_2$, and $S_2$ and the diurnal constituents, $K_1$, $O_1$, $P_1$, $Q_1$, and $S_1$, (Wollanksi, and Elliot, 2016). Other constituents that typically may contribute non-trivially to overall tidal amplitudes at the coast include higher-frequency non-linear shallow water "overtides" and annual and semiannual constituents (which are typically due mainly to seasonal oceanographic and meteorological variability, rather than astronomical forcing). The ROMS model is capable of dynamically reproducing both of these types contituents (at least to some degree).

Section2.3 the problem with introduction of tides in model adopting 360 day long year does not appear relevant because tides are not included in the climate change experiments (authors write this a few lines below)

Yes, we agree with the reviewer and removed this paragraph from manuscript.

Page 5 , lines 11-13, RMSE, STDE and correlation are weak metrics for validation of extremes in a time-series. High correlation and low RMSE can be obtained also if extremes are poorly reproduced. Further, to validate a model percent errors should be considered, particularly for extremes. To compare magnitude of the error to the magnitude of the observed value is important.

We are aiming to compare the analyses to that given by Haigh et al, 2014a. The aim thus was to show that the model captures atmospheric driven variability generally well. We asses extremes via qq plots in Figure 4.

Page 5, lines 4.It is not clear to me how is the seasonal variability component defined and computed in this study

We follow the methodology of Haigh et al,2014a. The seaosnal component is calculated by using a 30day running mean over the detided and detrended time series. This removes basically the high frequency (weather driven) variability. We changed the paragraph to make this clearer.

Page 5, line 31. If the 30-day running mean is subtracted to the signal, I expect that the steric contribution on the residual is small

We are not quite sure what the reviewer means. We use the 30 day running mean to tease out the seasonal signal. This follows the method of Haigh et al, 2014a. The line in question is discussing the findings of Forbes and Church 1983 who used measurements to show that the strong seasonal signal in sea level in the Gulf of Carpentaria comprises a barotropic component (northwest monsoon winds that produce wind setup in the gulf) and a steric component due to seasonal changes in temperature and salinity of the water column and these both have a maximum postive effect on sea levels in January. Our model is barotropic so can only capture the barotropic component of sea level variations.

*Sea level residuals.*
It is not clear to me what we learn from the considered examples. What have been the criteria for their selection

The examples have been selected to illustrate the main weather systems that cause storm surges along different coastal regions in Australia. The year 1997 was selected as it contained examples of extreme sea levels along each coastal region examined. We have reworded the relevant paragraphs to make this clearer .

Page 6, lines 33-34 to blame the inaccurate meteorological forcing is often correct, but it is also an easy way out. Can the authors provide an argument to support this?

Yes, this is true. Arguably there are a couple of reasons why the model is not able to reproduce SSH anomalies correctly (1) representation error of model and atmospheric forcing (i.e. grid resolution, bathymetry errors, coastlines not resolved properly, errors in the atmospheric forcing, limited temporal resolution), (2) model physics - a 2D model will only ever resolve the first barotropic mode of coastally trapped waves. Higher order modes may be necessary to properly account for all the variability.
We have added discussion on these points (see next point).

Page 7, line 13. It is not clear why in this specific location wave set up is expected to be a relevant contribution and could explain the underestimated sea level by the model. This should be explained in terms of location of the gauge and morphology of the sea bottom (including depth).

We agree with the reviewer and changed the paragraph. In fact what appears to happen here is explained by the following:

The missing peak may be explained by an insufficiently resolved modelled coastally trapped wave (CTW). Studies like Woodham et al, 2013 suggest speeds of CTW between 2-4m/s. CTW travel anticlockwise around Australia. It takes about 5-6 days for CTW to travel the distance from Port Kembla to Rosslyn Bay. On the 10th may a coastal low produced a surge in Port Kembla that excited a CTW. According to Woodham et al, CTW can cause sea level elevations of 0.25m which is in the order of what has been observed at Rosslyn Bay about 5-7 days later. The ROMS model does not capture this elevation,. This may potentially be due to its barotropic nature which does not allow higher order (bariclinic) modes of CTW to develop. Unresolved bathymetric features over the Great Barrier Reef are also candiates for explaining the model behaviour. We have amended the text to discuss these possible explanations.

Page 8 lines 6-11. To which figures do these sentences refer?
These sentences refer to Figure 6. We made appropriate changes in the text.

Authors consider the total sea level ZTM , Its tidal ZT and meteorological ZM components, all computed separately by independent simulation. Defining the residual ZR= ZTM - ZT, they find that peaks (ranks) of ZR and ZM agree and conclude that time-surge interaction is negligible. However, this is in contrast with the lack of agreement between ZTM and ZT + ZM , which shows that tides substantially decrease the importance of the meteorological contribution to sea level extremes. Therefore, to me it seems that tides are not relevant for computing correctly the maxima of the storm surge, but actual sea level maxima are affected (decreased) by tide-surge interaction (practically high tidal levels reduce the contribution of the surge to the maxima). Further, the whole analysis applies at the location of the tide gauge. I suspect that at the actual coastal line, at the shore, analysis can produce different results.

We agree with the assertion by the reviewer which is what we wrote in the manuscript. We have slightly revised the paragraph to make this clearer.

The statement that "climate models overall perform well" is too positive, considering the tendency of all simulations to underestimate high quantiles is some locations (fig.7).Such underestimate is particularly large for inmcm (note that the annotation in the figure is not consistent with the text which refers to this simulation as CC-I). Model simulations substantially underestimate extremes at several locations.

We have revised this paragraph to more explicitly describe the performance of the climate models. We have revised the figures to make naming of the climate models consistent with elsewhere in the paper.

*Seasonal mean maximum sea level change*

I find this part should be improved in several aspects 16.1) It discusses the multimodel mean at annual scale, and only individual models at seasonal scale.

The shown multimodel mean for the annual average is to compare results from CFSR forced surge model with observations and to show that the CMIP5 forced models show similar results in terms of overall distribution. The individual climate model results are broken into seasons to better understand the role that seasonal weather and circulation changes have on the results.

There is no indications whether changes are statistically significant for individual models. I suggest to mask in figure 11 (central panel) values when models do not agree on the sign of the change or add, anyway, an indication of the level of consensus among models.

We have modified the figure to have hatching where the multi-model ensemble is not in agreement on the sign of the change.

There is a discussion of the link of the observed changes of extremes with changes of wind speed. However, it is not clear why changes of mean speed are relevant for extremes and whether figure 10 is a multimodel mean or it represents the winds driving the CC-A simulations.

Yes, we agree with the reviewer and clarified this. Figure 10 demonstrates a possible mechanism that may explain the observed increase in extremes seen for ACCESS-R and HadGEM forced model simulations. Stronger mean monsoonal winds will cause more wind setup over the GoC. This in turn will increase the likelihood of extremes to happen.

**Reviewer 2**
Their conclusion is that tide-surge interaction is strong (at least at some parts along the coast) for individual extremes but may be neglected for statistics over longer periods. For me, this is not very conclusive as potential changes in tides are also not taken into account.

Yes – the author is correct we conclude 2 things in this section: (1) surges occur more often during low tide for some locations when tides are included thus leading to smaller total sea levels. This means the timing of the surge and tides is interlocked at times owing to non-linear interactions. However, we also show (2) that the height of the surge regardless of including tides or not does not change when looked at it in a ranked sense (or only to a small degree). We argue that because of point (2) we do not need to include tides in the future climate scenarios as our main interest is the effect of change in coastal surge driven by atmospheric forcing.

I understand somehow that changes in the met. forcing alone may be visible in a surge-only run and potential changes may (roughly) be inferred thereof but only if relative changes in he met induced component are investigated.

We are not sure what the reviewer is suggesting here.

Climate change will also affect the base water level (MSL) which has not been considered in your experiments, right? From SLR, the propagation of the surge will be affected influencing the timing (and hights) of surge events.

Yes we agree with the reviewer that e.g. SLR has the potential to change the speed of the gravity tidal wave and thereby also has the potential to change the distribution of tidal phases/ amplitutdes around the globe. However, in order to add such an effect into a surge model we would need to have access to newly generated tidal constituents (calculated by inverse modelling). Such a dataset is not available do date (to our knowledge). Also consider the large uncertainty in regional sea level rise projections that one needs to take into account. Furthermore changes due to SLR are in the order of meters which is small compared to errors/ uncertainty in the bathymetric datasets. I can see that one could change bottom topography to model such an effect to understand the sensitivity but this is beyond the scope of our study.

Furthermore, also the tidal propagation may/will change with SLR having the potential to further increase water levels and partly compensate for the "mostly" negative trend in ESL changes you reported stemming from the met. only approach

Yes, MSL and RSL both have the ability to increase current tidal propagation. To a first approximation components are often linearly added which leads to different exceedence probability thresholds. Note, however, that in our manuscript we only consider atmospheric driven changes in SSH. We added a paragraph discussion SLR scenarios.

The period of 20 yrs you consider for the future climate conditions are too short to draw robust conclusions. Usually a period of 30yrs is used to estimate changes in the met. forcing. Please consider extending your modelling or discuss why you chose this short period, how it affects your results.

The choice of twenty-year time slices was to align the hydrodynamic model output to wave model simulations carried out using the same climate models over the same time period that was published in Hemer and Trenham (2016). Our aim was to be able to couple hydrodynamic extremes with wave-induced extremes (e.g. wave setup or runup) in future work. We acknowledge that 20 years may be too short to assess the role of future changes to interannual variability (i.e. ENSO) on weather events that cause extreme sea levels such as tropical cyclones, but as we already note, the GCMs do not adequately resolve TCs anyway so the focus of our study is on the contribution of large scale circulation changes to extreme sea levels. We feel that 20-year time slices are adequate for assessing how large scale circulation changes will affect drivers of sea levels around much of Australia's coast where seasonally varying weather systems are a major cause of extreme sea levels.

(Hemer, M. A. and C. E. Trenham, 2016: Evaluation of a CMIP5 derived dynamical global wind wave climate model ensemble. Ocean Modelling, 103, 190-203, doi:https://doi.org/10.1016/j.ocemod.2015.10.009.)

page 3, line 30: 1' x 1', x missing
Fixed that page 7, line 26: Due to computational constraints, we demonstrate that... From my point of view, this is not a good argumentation
Yes we agree and changed this page 9, line 8: BM covers 1981-2012, right? so the common period is '81-'99. Should be clear
This has been fixed page 9, line 18: Albany not shown in Figure 1
The reference to Albany in the text should have been Esperance. This has been fixed

Fig1: Could be helpful to show the average tidal range (e.g. based on TPXO) over the entire area

Fig.2: Please define the dots (semi- and diurnal)
We explain the dots in the figure caption.

Fig.4: All R2s show values of ~1. This is a bit misleading, as most stations over- and/or underestimate the extremes. Also the R2 is not mentioned
We agree and have removed the R2 values.

Fig. 6: Units missing; please highlight meaning of surge and residual again; for me, the figure shows a clear tide-surge interaction which cannot be neglected. Also for the largest events as e.g. in Rosslyn Bay or Darwin
Units have been added. More information has been added to the figure caption

Fig. 8: Portland not given in the Fig. , what is happening at the northern part (Milner Bay)
Have added Portland. We assume the reviewer is referring to the large difference between the Milner Bay observations and model results. They may be explained by the fact that the Milner Bay tide gauge is located at the south side of Groote Island which is not very well resolved in the model.

All figures would benefit from detailed captions.

We have added more information to the figure captions as necessary.

**Atmospheric Circulation Changes and their Impact on Extreme Sea Levels around Australia**

Frank Colberg[1], Kathleen L. McInnes[2], Julian O'Grady[2] and Ron Hoeke[2]

[1]Bureau of Meteorology,
f.colberg@bom.gov.au

[2]Climate Science Centre,
CSIRO Marine and Atmospheric Research,
Aspendale, 3195, Australia
kathleen.mcinnes@csiro.au
julian.ogrady@csiro.au
ron.hoeke@csiro.au

Revised November 2018

Journal: Natural Hazards and Earth System Sciences (NHESS)

| Deleted: For Submission: Jan 2017 |

| Deleted: JGR, Ocean Dynamics, ???? Ocean Modelling??? IJC ??? |

[revised manuscript text omitted]

---

## Author Response (AR2)

**Response to the reviewer**

The motivation for the 20-year length of the analyzed period.
Page 5, line 9-14 Authors write "The twenty-year time slices were chosen to align the hydrodynamic model output to wave model simulations described in Hemer and Trenham [2016] with the aim to combine hydrodynamic extremes with wave-induced extremes (e.g. wave setup or runup) in future work."
To me this looks a weak justification: having a longer period would not have prevented this planned future work
We changed the text significantly, both in the section 2.3 indicated by the reviewer, and in the concluding section to make justification clearer, with appropriate caveats.

And further "The 20-year time slices are deemed adequate for assessing how large scale circulation changes will affect the drivers of ESLs around much of the Australian coast where seasonally varying weather systems are a major cause of extreme sea levels." I think this is also weak as extreme weather frequency is affected by multidecadal variability. Could references be added to support that "20-year time slices are deemed adequate"?
See previous, response – also we have expanded reasoning about where (spatially) uncertainties may be high/low.

Neglecting of the tide-surge component, is acceptable because their computation would have been expensive, and benefits on climate change detection arguable, however:
Page 5 at line 20-21 authors write " [tide-surge interaction] may impact substantially on an individual surge event, [but] it does not change the surge statistics over a period of years to decades" . If it impacts significantly individual events, why it has no effect on the statistics? Even though it will not change the mean surge level, it would make the distribution broader or narrower, therefore eventually, increasing ordecreasing extreme values
Yes, we changed the sentence to make this clearer. The statistics don't change dramatically for most of Australias's coastline. Some changes are observed for selected stations such as Broome, Darwin and Burnie where maximum surges tend to be systematically underestimated/ overestimated for ranges of maximum SLA (Figure 6).

Further, (figure 4) extreme surges are underestimated in the model simulation and this might explain why tide-surge interaction is weak
This is a possibility.

At Line 1-2 at page 8 "some evidence points towards this peak being generated by a coastally trapped wave (CTW)." ...It is not clear what is the evidence.
The evidence is discussed later in the paragraph. We added a notion in the sentence to make it more clear.

At line 17 page 3
The conclusions Lionello et al are not reported fully (just read the last sentence in the abstract of that paper). The sentence ", but mass addition will likely determine an increase of the water level maxima" should be added
OK – we extended the sentence as suggested by the reviewer.

**Atmospheric Circulation Changes and their Impact on Extreme Sea Levels around Australia**

Frank Colberg[1], Kathleen L. McInnes[2], Julian O'Grady[2] and Ron Hoeke[2]

[1]Bureau of Meteorology,
f.colberg@bom.gov.au

[2]Climate Science Centre,
CSIRO Marine and Atmospheric Research,
Aspendale, 3195, Australia
kathleen.mcinnes@csiro.au
julian.ogrady@csiro.au
ron.hoeke@csiro.au

Revised November 2018

Journal: Natural Hazards and Earth System Sciences (NHESS)

[revised manuscript text omitted]

---

## Author Response (AR3)

Dear Editor(s),

As requested, we have addressed the following technical corrections:
Line 134: I guess a "the" is missing before "mass addition component"
Lines 219-221: the sentence reads a bit awkward, please enhance
Lines 225-228: Please separate into two sentences.
Line 556: I'd delete the text "(such as the 20-year time slices used here)", not needed

Please let me know if you need any other corrections, materials or information.

Cheers,
Ron